# Functional anatomy of zinc finger antiviral protein complexes

Jennifer A. Bohn[1], Jennifer L. Meagher[2], Matthew A. Takata[1], Daniel Gonçalves-Carneiro[1], Zoe C. Yeoh[2,3], Melanie D. Ohi [2,4], Janet L. Smith [2,3] & Paul D. Bieniasz [1,5] ✉

ZAP is an antiviral protein that binds to and depletes viral RNA, which is often distinguished from vertebrate host RNA by its elevated CpG content. Two ZAP cofactors, TRIM25 and KHNYN, have activities that are poorly understood. Here, we show that functional interactions between ZAP, TRIM25 and KHNYN involve multiple domains of each protein, and that the ability of TRIM25 to multimerize via its RING domain augments ZAP activity and specificity. We show that KHNYN is an active nuclease that acts in a partly redundant manner with its homolog N4BP1. The ZAP N-terminal RNA binding domain constitutes a minimal core that is essential for antiviral complex activity, and we present a crystal structure of this domain that reveals contacts with the functionally required KHNYN C-terminal domain. These contacts are remote from the ZAP CpG binding site and would not interfere with RNA binding. Based on our dissection of ZAP, TRIM25 and KHNYN functional anatomy, we could design artificial chimeric antiviral proteins that reconstitute the antiviral function of the intact authentic proteins, but in the absence of protein domains that are otherwise required for activity. Together, these results suggest a model for the RNA recognition and action of ZAP-containing antiviral protein complexes.

Zinc-finger (ZnF)-containing proteins play vital roles in immune responses during viral infection. While some ZnF proteins selectively bind host or viral RNA, others regulate levels of immune signaling molecules such as cytokines[1,2]. The zinc-finger antiviral protein (ZAP) directly binds to viral RNA, resulting in its destruction and virus attenuation[3–6]. ZAP specifically binds to CpG dinucleotides, which have been depleted from vertebrate genomes over evolutionary time, enabling ZAP to distinguish self from non-self RNAs[7]. The ZAP N-terminus comprises the RNA binding domain (ZAP RBD), which includes four CCCH-type ZnFs, with CpG-specific binding largely attributed to ZnF2[8]. ZAP is naturally found mainly as two isoforms, termed ZAP-S (short) and ZAP-L (long), resulting from alternative splicing[9]. In addition to the RBD, ZAP-S, and ZAP-L contain a central region with a fifth ZnF of unknown function and two canonical WWE domains, which have poly-(ADP-ribose)-binding activity that enhances

antiviral activity[10]. ZAP-L contains an additional C-terminal poly-(ADP-ribose) polymerase (PARP)-like domain that lacks catalytic activity and a C-terminal S-farnesylation motif[11–13].

Of the multiple ZAP cofactors that have been reported, tripartite motif protein 25 (TRIM25) and the KH-like and NYN domain-containing protein (KHNYN) appear especially important for antiviral activity[14–18]. TRIM25 is an E3-ubiquitin ligase with an N-terminal RING domain, a B-box, a coiled-coil, and a C-terminal SPRY domain and is one of numerous proteins with similar domain architecture[19]. Many TRIM proteins form higher-order oligomers via the RING domain[20,21], and in some cases, this oligomerization is critical for antiviral activity[22,23] or, in the case of TRIM25, for RIG-I activation[24]. TRIM25 ubiquitin ligase activity is reported to be important for inhibition of Sindbis virus translation[25]. Previous work suggests that the SPRY domain is critical for TRIM25/ZAP interactions, but the role of the TRIM25 oligomeric

[1]Laboratory of Retrovirology, The Rockefeller University, New York, NY 10065, USA. [2]Life Sciences Institute, University of Michigan, Ann Arbor, MI 48109, USA. [3]Dept Biological Chemistry, University of Michigan, Ann Arbor, MI 48109, USA. [4]Dept Cell & Developmental Biology, University of Michigan, Ann Arbor, MI 48109, USA. [5]Howard Hughes Medical Institute, The Rockefeller University, New York, NY 10065, USA. ✉e-mail: pbieniasz@rockefeller.edu

state, ubiquitin ligase activity and mechanistic details of how TRIM25 enables ZAP function remain elusive[16,26].

KHNYN is a putative endonuclease shown to interact with ZAP and TRIM25[18]. While canonical KH-domains bind RNA, it is not known whether the N-terminal KH domain of KHNYN serves this function and, if so, whether such RNA binding activity contributes to ZAP activity. KHNYN also has a small C-terminal domain that is predicted to bind NEDD8, a ubiquitin-like molecule; however, again, it is unknown whether this predicted interaction plays a role in ZAP antiviral complex assembly or activity[27,28]. Additionally, while the NYN domain of KHNYN is a sequence homolog of the MCPIP1 NYN domain, which has exonuclease activity, it has not been shown that KHNYN is a nuclease[29].

Thus, while TRIM25 and KHNYN may act in concert with ZAP to exert antiviral activity, there is little mechanistic understanding of how putative antiviral complexes that recognize and degrade CpG-rich RNA are assembled. Herein, we describe the functional roles for each component of the ZAP antiviral complex and show that assembly of an antiviral complex is driven by an extensive set of protein−protein interactions, that together result in specific targeting of CpG-rich RNA substrates. We also show that KHNYN is an active RNA endonuclease and that another NYN domain family member, N4BP1, can substitute for KHNYN in ZAP-dependent antiviral activity. We further describe a crystal structure that captures one of the interactions between ZAP and KHNYN, which contributes to antiviral complex formation. Finally, we derive an overall model for the organization of antiviral complexes containing ZAP, TRIM25, and KHNYN and use that information to design active artificial chimeric antiviral proteins composed of individual domains of ZAP, KHNYN, or TRIM25.

## Results

### Cofactor requirement and redundant role of KHNYN and N4BP1 in ZAP antiviral function

To assess the requirement for TRIM25 and KHNYN in ZAP-dependent antiviral activity against CpG-enriched HIV-1, we compared the yield of infectious wild-type virus ($HIV-1_{WT}$) and a synonymously mutated CpG-enriched derivative ($HIV-1_{CG}$)[7] by infecting Hela cells lacking one of each of these proteins with varying doses of pseudotyped virus and measuring infectious virus yield over a single cycle of replication (Fig. 1a−e and Supplementary Fig. 1a). CRISPR knockout of each gene/protein was validated by western blot analysis or by the presence of frameshift mutations at the targeted locus (examples are shown in Supplementary Fig. 1b, c). In control cells transduced with an empty CRISPR vector, a single cycle of $HIV-1_{CG}$ replication was attenuated, yielding 10−100-fold less infectious virus compared to $HIV-1_{WT}$ (Fig. 1a−e). While $HIV-1_{CG}$ replication was completely restored in $ZAP^{-/-}$ cells (Fig. 1a, b) it was not completely restored by knockout of TRIM25 or KHNYN (Fig. 1c, d), leading us to consider the possibility that other cofactors are yet to be identified. The most closely related protein to KHNYN in human cells is NEDD4 binding protein 1 (N4BP1), with which it shares ~37% sequence identity. We generated $N4BP1^{-/-}$ HeLa cells and found that $HIV-1_{CG}$ replication remained at least partly attenuated (Fig. 1e). However, we also generated $KHNYN^{-/-}/N4BP1^{-/-}$ Hela cells and found that this double knockout restored $HIV-1_{CG}$ replication to near $HIV-1_{WT}$ levels, suggesting a redundant role for KHNYN and N4BP1 in ZAP antiviral function (Fig. 1f). Consistent with this conclusion, reconstitution of the $KHNYN^{-/-}/N4BP1^{-/-}$ cells by expression of either KHNYN or N4BP1 was sufficient to specifically attenuate $HIV-1_{CG}$ replication (Fig. 1g, h).

To determine whether these findings could be recapitulated using an independent virus system, we compared the replication of wild-type human enterovirus A71 ($EV71_{WT}$) with an EV71 mutant that was recoded to contain a genome segment with CpG dinucleotides in an optimal context for ZAP inhibition ($EV71_{CG}$)[30]. In control cells, $EV71_{CG}$ replicated to levels that were 100-fold lower than $EV71_{WT}$, and this CpG-imposed attenuation was completely absent in $ZAP^{-/-}$ cells

(Fig. 1i). Knockout of TRIM25 resulted in partial restoration of $EV71_{CG}$ replication (-10-fold). Knockout of KHNYN or N4BP1 individually had little effect on $EV71_{CG}$ attenuation, but knockout of both substantially rescued $EV71_{CG}$ replication, although some residual attenuation remained in the $KHNYN^{-/-}/N4BP1^{-/-}$ double knockout cells (Fig. 1i). Overall. these data confirm an activity-enhancing role for TRIM25 in ZAP antiviral activity[16,17], as well as a role for KHNYN[18]. However, they also uncover a similar, but redundant to KHNYN, role for N4BP1 in ZAP-dependent antiviral activity. We speculate that the residual replication deficiency exhibited by $HIV-1_{CG}$ and $EV71_{CG}$ in $KHNYN^{-/-}/N4BP1^{-/-}$ Hela cells that is absent in $ZAP^{-/-}$ cells may be due to the participation of additional nucleases of the NYN family, or other families[14].

### An experimental system for reconstitution and analysis of ZAP, TRIM25, and KHNYN function

To probe the molecular details of the processes that drive the formation of active ZAP antiviral complexes, we used CRISPR/Cas9 to engineer a HEK293T-derived $ZAP^{-/-}$ $TRIM25^{-/-}$ cell line[16] such that it lacked any of the aforementioned known cofactors ($ZAP^{-/-}$ $TRIM25^{-/-}$ $KHNYN^{-/-}$ $N4BP1^{-/-}$) (Supplementary Fig. 1c). Thus, any combination of the aforementioned proteins contributing to ZAP function could be reconstituted by transient transfection (Fig. 1j). We assessed which components were necessary or sufficient for specific attenuation of $HIV-1_{CG}$ by co-transfecting combinations of expression plasmids alongside proviral plasmids encoding $HIV-1_{WT}$ or $HIV-1_{CG}$ into $ZAP^{-/-}$ $TRIM25^{-/-}$ $KHNYN^{-/-}$ $N4BP1^{-/-}$ HEK293T cells and measuring infectious virus yield. To validate this approach, we first tested the ability of varying levels of KHNYN or a KHNYN mutant with Ala substitutions for Asp residues at the putative active site ($KHNYN_{cat}$, see methods) to attenuate $HIV-1_{WT}$ or $HIV-1_{CG}$. Transfection of plasmids expressing KHNYN or $KHNYN_{cat}$ alone did not reduce the yield of either $HIV-1_{WT}$ or $HIV-1_{CG}$ (Fig. 1k) nor did a combination of plasmids encoding KHNYN and TRIM25 (Fig. 1l). Co-transfection of plasmids encoding ZAP and KHNYN (in the absence of TRIM25) resulted in modest reduction in $HIV-1_{CG}$ yield, but this combination also had weak activity against $HIV-1_{WT}$ (Fig. 1m). Co-transfection of plasmids expressing ZAP, KHNYN and TRIM25 resulted in potent antiviral activity (100-fold reduction in infectious virus yield) that was specific for $HIV-1_{CG}$, entirely absent for $HIV-1_{WT}$, and dependent on KHNYN putative catalytic site amino acids (Fig. 1n). Thus, the presence of TRIM25 enhanced both the activity and the $HIV-1_{CG}$ specificity of the antiviral activity exhibited by the ZAP/KHNYN combination (Fig. 1m, n). No antiviral activity was observed when plasmids expressing ZAP or TRIM25 were transfected alone or together in the absence of KHNYN (Supplementary Fig. 1d). Overall, these results confirm that ZAP and KHNYN are required for reconstitution of antiviral activity in $ZAP^{-/-}$ $TRIM25^{-/-}$ $KHNYN^{-/-}$ $N4BP1^{-/-}$ cells, and further show that the presence of TRIM25 enhances both overall antiviral activity and CpG-specificity in this context.

### ZAP self-associates, and its N-terminal domain functionally interacts with both KHNYN and TRIM25

Using this reconstitution approach, we next sought to determine the domain requirements for each component (ZAP, TRIM25, and KHNYN) in $HIV-1_{CG}$-specific antiviral activity. First, we asked which domains of ZAP are necessary and sufficient for antiviral activity. Plasmids expressing TRIM25, KHNYN and ZAP fragments of varying length that included only the RBD (ZAP-X with amino acids 1−227 and ZAP-N with amino acids 1−254), the RBD and a central putatively disordered domain (ZAP-D with amino acids 1−511), or the naturally occurring isoforms (ZAP-S and ZAP-L) were transfected with $HIV-1_{WT}$ or $HIV-1_{CG}$ proviral plasmids into $ZAP^{-/-}$ $TRIM25^{-/-}$ $KHNYN^{-/-}$ $N4BP1^{-/-}$ HEK293T cells and infectious virus production was measured (Fig. 2a). The ZAP RBD-only fragments (ZAP-X and ZAP-N) exerted antiviral activity when coexpressed with KHNYN in the presence of TRIM25 and to a greatly reduced degree in its absence (Fig. 2b, c). In these

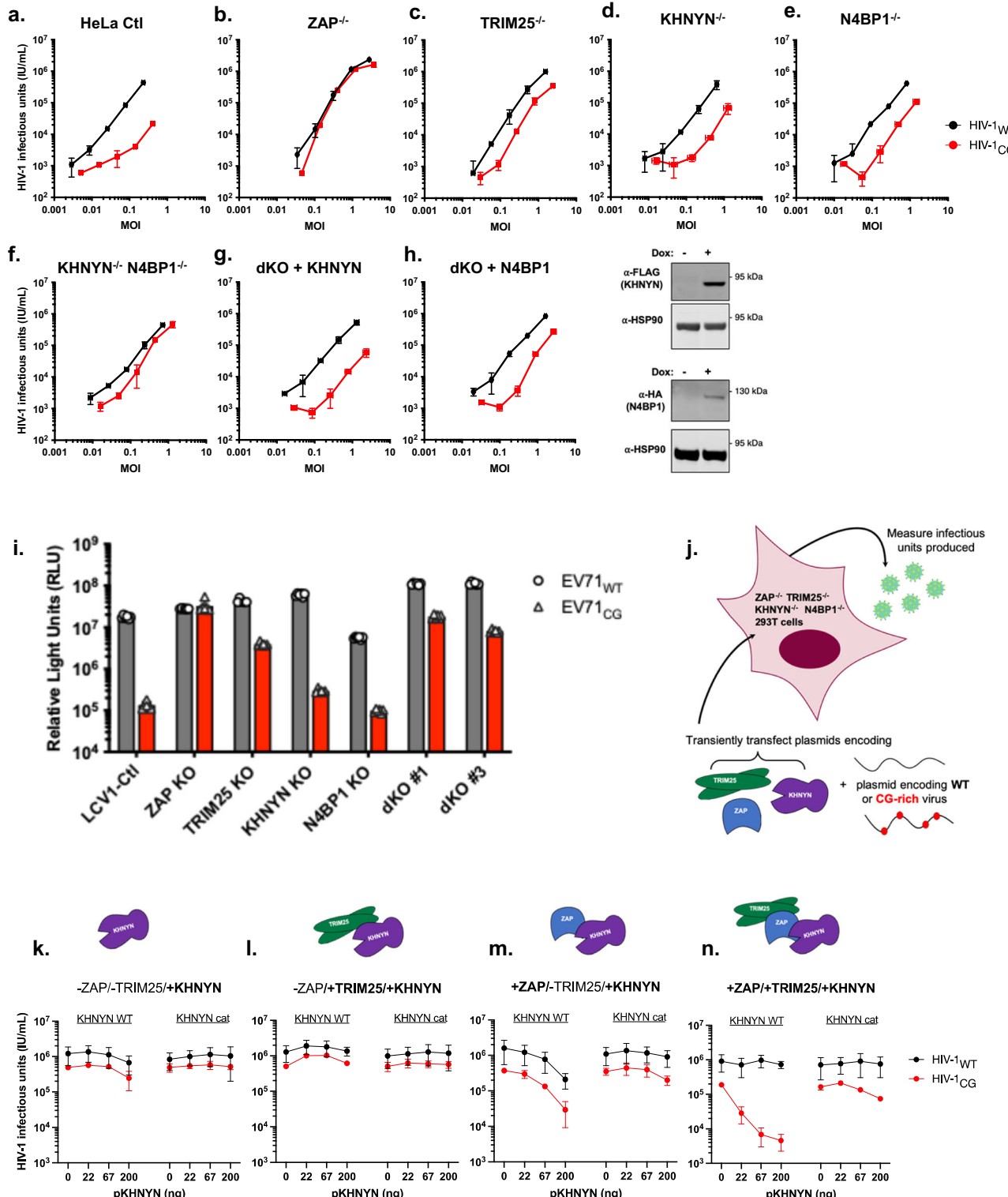

reconstitution experiments, some nonspecific antiviral activity against HIV-1$_{WT}$ was noted as ZAP expression was increased. This property was especially evident with the truncated ZAP proteins and in the absence of TRIM25, but was nearly absent when full-length ZAP, TRIM25, and KHNYN were coexpressed (Fig. 2b, c). Increasing ZAP overexpression also influenced the levels of KHNYN and TRIM25 even though a fixed amount of KHNYN and TRIM25 expression plasmids were co-transfected. For example, KHNYN protein levels increased as ZAP expression was increased (Supplementary Fig. 2). These findings are consistent with a model where ZAP, TRIM25, and KHNYN form

functional complexes, with the ZAP N-terminal domain minimally required for activity.

We next assessed which domains of ZAP were required for association with itself, with KHNYN, or with TRIM25 using co-immunoprecipitation assays in cells that were stably transduced with doxycycline (dox)-inducible C-terminally FLAG-tagged ZAP-N or ZAP-L. To evaluate ZAP self-association, co-immunoprecipitations were conducted with these cells transiently transfected with plasmids expressing various HA-tagged truncated ZAP proteins. Importantly, cell lysates were RNase A treated prior to immunoprecipitation to exclude

**Fig. 1 | Proteins required for ZAP antiviral activity. a** Single-cycle HIV-1$_{WT}$ (black) and HIV-1$_{CG}$ (red), infection of HeLa control cells (transduced with empty CRISPR-no guide- vector) multiplicity of infection (MOI) for input virus is plotted on the x-axis and subsequent virus yield from those infected cells (infectious units/mL) is plotted on the y-axis. Data were reported as mean ± standard error of the mean (sem) of three biological replicates. **b–f** Single-cycle infection of ZAP$^{-/-}$ (**b**) TRIM25$^{-/-}$ (**c**) KHNYN$^{-/-}$ (**d**) N4BP1$^{-/-}$ (**e**) KHNYN$^{-/-}$ N4BP1$^{-/-}$ (**f**) Hela cells, as in (**a**). **g, h** Single-cycle infection of KHNYN$^{-/-}$ N4BP1$^{-/-}$ Hela cells as in (**a**) following reconstitution with a doxycycline-inducible KHNYN-FLAG (**g**) or N4BP1-3xHA (**h**) expression vector. Expression of KHNYN-FLAG or N4BP1-3xHA in the presence or absence of Dox is shown on the right. **i** EV71$_{WT}$ (dark gray bars) vs EV71$_{CG}$ (red bars) replication in the HeLa cell lines as in (**a–f**) above. Luminescence (relative light units) generated by the NanoLuc reporter EV71 viruses measured at 48 h after infection is shown and

error bars represent six biological replicates. **j** Cartoon schematic representation of HEK293T ZAP$^{-/-}$/TRIM25$^{-/-}$/KHNYN$^{-/-}$/N4BP1$^{-/-}$ cell transfection-based reconstitution system to measure virus yield in the presence or absence of ZAP antiviral complex components. **k–n** Reconstitution of ZAP, TRIM25, and KHNYN antiviral activity. Increasing amounts (ng) of plasmids expressing KHNYN$_{WT}$ or KHNYN$_{cat}$ (putative catalytic site mutant) were transfected with 350 ng HIV-1$_{WT}$ or HIV-1$_{CG}$ proviral plasmids along with empty vector (**k**), 75 ng TRIM25-expressing plasmid (**l**), 150 ng ZAP-expressing plasmid (**m**), or both 150 ng ZAP- and 75 ng TRIM25-expression plasmids (**n**) Infectious virus yield (in infectious units/mL) from transfected cells measured by infection of MT4 LTR-GFP cells and flow cytometry. Data were reported as mean ± sem of two biological replicates and are representative of three independent experiments.

the possibility that RNA drove protein–protein associations. In these experiments, ZAP-L robustly and specifically co-immunoprecipitated all truncated ZAP proteins tested, including ZAP fragments with no shared domains (ZAP-N [1–254] and ZAP-L$_{\Delta RBD}$ [255-902]) (Fig. 2d). In contrast to stably expressed ZAP-L, the stably expressed ZAP-N poorly co-immunoprecipitated each of the ZAP fragments (Fig. 2d). The efficiency with which stably expressed ZAP-L-FLAG coprecipitated transiently expressed ZAP-N-HA was greater than the efficiency with which ZAP-N-FLAG coprecipitated ZAP-L-HA. This is likely because the ZAP-L-FLAG and ZAP-N-FLAG "bait" proteins were expressed at significantly lower levels than the transiently expressed ZAP-N-HA and ZAP-L-HA "prey" proteins. Because ZAP-L homodimers assemble more efficiently than ZAP-N/ZAP-L heterodimers or ZAP-N/ZAP-N homodimers, then excess of ZAP-N would be required to form ZAP-N/ZAP-L heterodimers if both ZAP-L and ZAP-N proteins are present. Overall, these data suggest that ZAP self-associates in a manner that involves multiple ZAP domains.

We performed similar co-immunoprecipitation experiments in ZAP$^{-/-}$ TRIM25$^{-/-}$ KHNYN$^{-/-}$ N4BP1$^{-/-}$ HEK293T cells stably transduced with a dox-inducible C-terminally FLAG-tagged KHNYN expression construct and transiently transfected with full length or truncated HA-tagged ZAP expression constructs (Fig. 2e). All ZAP variant proteins containing the N-terminal RBD co-immunoprecipitated with KHNYN, while a truncated ZAP protein lacking the RBD (ZAP-L$_{\Delta RBD}$) did not (Fig. 2e). This result suggests that KHNYN binds to the ZAP RBD. Similar approaches in which endogenous TRIM25 was immunoprecipitated from 293T cells transfected with plasmids expressing truncated ZAP proteins indicated that TRIM25 association with ZAP-X and ZAP-N, was weak, but TRIM25 efficiently coprecipitated ZAP-D, ZAP-S, and ZAP-L (Fig. 2f). A ZAP mutant that lacked RNA binding activity (ZAP-X RNA$_{null}$, (R74A, R75A, K76A))[26] also efficiently coprecipitated TRIM25, but ZAP-L$_{\Delta RBD}$ which lack residues 1–254 did not (Fig. 2f). These results suggest that the N-terminal RBD, but not RNA binding activity itself, is also important for ZAP association with TRIM25. Further, ZAP interaction with TRIM25 is facilitated by additional ZAP domains outside the RBD that may also drive ZAP multimerization.

Overall, these functional reconstitution and co-immunoprecipitation experiments support a model where the N-terminal RBD is central in nucleating assembly of functional antiviral complexes containing ZAP, while additional ZAP domains facilitate self-association, TRIM25 recruitment and also impart greater CpG-specificity on the antiviral action of ZAP (Fig. 2d).

**KHNYN domains required for ZAP-dependent antiviral activity**
Next, we sought to determine which domains of KHNYN are required for antiviral activity. KHNYN is composed of annotated domains that include a K-homology (KH)-domain, a protein domain that typically binds RNA[31], an N4BP1- and YacP-like nuclease (NYN)-like domain[29], as well as a cullin-binding domain associating with NEDD8 (CUBAN) domain[27], referred to here as the CTD (C-terminal domain) (Fig. 3a). Additional sequences located between the KH and NYN domains have

no similarity to proteins of known function. We generated a series of KHNYN deletion constructs and compared the antiviral activities of each of these constructs to KHNYN$_{WT}$ and KHNYN$_{cat}$ in the ZAP$^{-/-}$ TRIM25$^{-/-}$ KHNYN$^{-/-}$ N4BP1$^{-/-}$ HEK293T cell reconstitution assay. Expression levels of the truncated KHNYN proteins were similar to the full-length protein, except for KHNYN$_{\Delta KH}$ which showed reduced protein levels, possibly contributing to the loss of antiviral activity observed for this protein (Supplementary Fig. 3a, b).

In the presence of both ZAP and TRIM25, deletion of either the KH or the CTD domain in KHNYN resulted in a 10-fold loss of antiviral activity against HIV-1$_{CG}$ as well a reduction in specificity for HIV-1$_{CG}$ versus HIV-1$_{WT}$ (Fig. 3b, left panel). Minimal antiviral activity was observed for the NYN-CTD construct, and no antiviral activity was observed when the isolated NYN catalytic domain was coexpressed with ZAP and TRIM25 (Fig. 3b, right panel). In reconstitution experiments that included ZAP but not TRIM25, KHNYN$_{WT}$ had moderate activity that was not completely specific for HIV-1$_{CG}$, while no activity was observed in the absence of TRIM25 for any of the deletion constructs (Fig. 3c left panel). These results suggest that functional interactions between KHNYN and ZAP can occur in the absence of TRIM25, but that TRIM25 augments the antiviral activity and specificity. Moreover, full-length KHNYN is required for full antiviral activity, and each of its non-catalytic domains contributes to the activity of ZAP antiviral complexes.

**Full-length KHNYN is required for assembly with ZAP and TRIM25**
For co-immunoprecipitation experiments, C-terminally FLAG-tagged KHNYN constructs were transfected into HEK293T cells expressing endogenous TRIM25. Endogenous TRIM25 co-immunoprecipitated with full-length KHNYN$_{WT}$-FLAG, KHNYN$_{\Delta KH}$-FLAG or KHNYN$_{\Delta CTD}$-FLAG (Fig. 3d), consistent with results in the reconstitution experiments in which KHNYN$_{\Delta KH}$ or KHNYN$_{\Delta CTD}$ gained activity in the presence of TRIM25 (Fig. 3b vs 3c left panels). The KH domain alone was able to co-immunoprecipitate reduced amounts of TRIM25 compared to KHNYN$_{WT}$, KHNYN$_{\Delta KH}$, or KHNYN$_{\Delta CTD}$ (Fig. 3d). No TRIM25 was observed to co-immunoprecipitate with KHNYN$_{NYN-CTD}$ or KHNYN$_{NYN}$ constructs. These results indicate that multiple KHNYN domains, in the N-terminal portions of the protein contribute to the interaction between KHNYN and TRIM25 (Fig. 3d).

We also tested which domains of KHNYN were required to co-immunoprecipitate ZAP-L in ZAP$^{-/-}$ TRIM25$^{-/-}$ KHNYN$^{-/-}$ N4BP1$^{-/-}$ HEK293T cells stably harboring a dox-inducible ZAP-L expression construct and transiently transfected with the suite of C-terminally FLAG-tagged KHNYN constructs. KHNYN$_{WT}$ and KHNYN$_{cat}$ both immunoprecipitated ZAP-L, as did the KHNYN$_{\Delta CTD}$ construct (Fig. 3e). KHNYN$_{\Delta KH}$ poorly co-immunoprecipitated ZAP-L, likely due to reduced protein expression. Notably the isolated KHNYN$_{KH}$ domain co-immunoprecipitated with ZAP-L, while KHNYN$_{NYN-CTD}$ did not (Fig. 3e). Overall, these results suggest that each KHNYN sub-domain contributes to antiviral activity, with the non-enzymatic KH and CTD

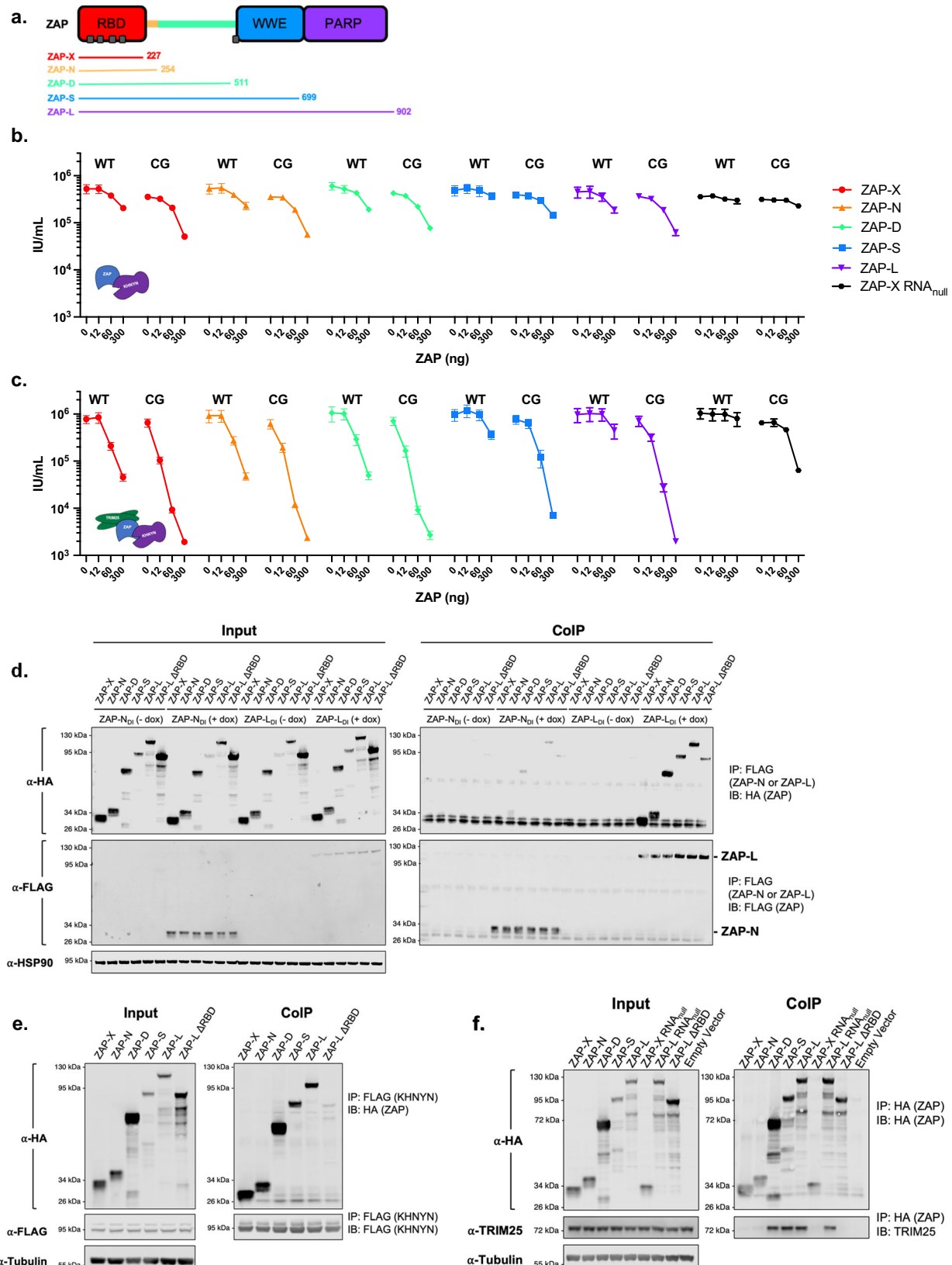

domains likely participating in protein–protein interactions with ZAP and/or TRIM25 that enable assembly of fully active antiviral complexes.

To confirm and extend the aforementioned results, we also tested whether tagged KHNYN could simultaneously co-immunoprecipitate endogenous ZAP-L and endogenous TRIM25 in 293T cells lacking KHNYN and N4BP1, and transfected with FLAG-tagged KHNYN (Supplementary Fig. 3c). Both endogenous TRIM25 and endogenous ZAP-L co-immunoprecipitated with full-length KHNYN$_{WT}$-FLAG, and KHNYN$_{\Delta CTD}$-FLAG. While TRIM25 but only trace amounts of ZAP-L coprecipitated with KHNYN$_{\Delta KH}$-FLAG. Together these results suggest the formation of a ternary complex containing ZAP-L, TRIM25, and

**Fig. 2 | ZAP domain requirement for CG-specific antiviral activity. a** Schematic depicting ZAP domain organization and truncated proteins used for reconstitution experiments, noting C-terminal domain boundaries. ZAP has an N-terminal RNA-binding domain (RBD, red) containing four zinc fingers (ZnFs), a putatively disordered domain (green), a fifth ZnF, WWE (blue), and PARP (purple) domains are at the C-terminus. **b** Reconstitution experiments in which increasing amounts of plasmids (ng) expressing the truncated ZAP expression plasmids shown in (**a**) were co-transfected with a constant amount (75 ng) of KHNYN expression plasmid in 293T ZAP$^{-/-}$ TRIM25$^{-/-}$ KHNYN$^{-/-}$ N4BP1$^{-/-}$ cells. Effects on HIV-1$_{WT}$ and HIV-1$_{CG}$ virus yield was measured as infectious units (IU/ml). Data were reported as mean ± sem of $n = 4$ biological replicates. **c** Same as (**b**) except that 75 ng of a TRIM25 expression plasmid was included in each transfection. **d** Co-immunoprecipitation of C-terminally 3xHA-tagged ZAP proteins (shown in (**a**)) transiently transfected in 293T ZAP$^{-/-}$ TRIM25$^{-/-}$ KHNYN$^{-/-}$ N4BP1$^{-/-}$ cells, with or without doxycycline-induced expression of ZAP-N-FLAG or ZAP-L-FLAG. Proteins were immunoprecipitated from cell lysates with anti-FLAG antibodies and subjected to western blot analysis. ZAP-L with the RNA-binding domain deleted (ZAP-L ΔRBD) was used to evaluate the role of the RBD in ZAP self-association. IP immunoprecipitation, IB immunoblot. **e** Co-immunoprecipitation of C-terminally 3xHA-tagged ZAP proteins (shown in (**a**)) transiently transfected in 293T ZAP$^{-/-}$ TRIM25$^{-/-}$ KHNYN$^{-/-}$ N4BP1$^{-/-}$ cells, with doxycycline-induced expression of KHNYN-FLAG. Proteins were immunoprecipitated from cell lysates with anti-FLAG antibodies and subjected to western blot analysis. **f** Co-immunoprecipitation of TRIM25 with C-terminally 3xHA-tagged ZAP proteins (shown in (**a**)) transiently transfected in 293T ZAP$^{-/-}$ TRIM25$^{-/-}$ KHNYN$^{-/-}$ N4BP1$^{-/-}$ cells. Proteins were immunoprecipitated from cell lysates with anti-HA antibody and subjected to western blot analyses with the indicated antibodies.

KHNYN and that the KHNYN KH domain is important for the formation of this putative complex.

## KHNYN is an active RNA endonuclease

While KHNYN has a protein domain that shares homology with a reported nuclease, bona fide nuclease activity has not been demonstrated. We expressed and purified FLAG-tagged KHNYN$_{WT}$ and KHNYN$_{cat}$ in KHNYN$^{-/-}$ N4BP1$^{-/-}$ dKO HEK293T cells (Supplementary Fig. 4a). The isolated protein was sufficient in purity to be quantified and utilized for enzyme assays (Supplementary Fig. 4a). We tested KHNYN activity on two 5'-fluorophore-labeled RNA substrates based on a high-affinity ZAP binding site in the HIV-1$_{CG}$ genome[7,8]. RNA substrate CG-5 is a 58-mer with five CG dinucleotides, and RNA substrate CG-0 is a 56-mer derivative of CG-5 in which each CG dinucleotide was replaced with AA, allowing us to also test whether substrate CpG dinucleotides are important for any observed KHNYN RNA nuclease activity (Fig. 4a). KHNYN$_{WT}$ cleaved both CG-5 and CG-0 substrates, causing accumulation of smaller RNA products over 4 h (Fig. 4b, c). The CG-5 substrate was not digested more rapidly than the CG-0 substrate (Fig. 4b, c), and neither CG-0 nor CG-5 was digested in control reactions in the absence of KHNYN or in the presence of KHNYN$_{cat}$ (Fig. 4b, c). We conclude that KHNYN is indeed an active nuclease in vitro, that nucleolytic and antiviral activity require the putative active site residues, but other components of the antiviral complex are required for CpG-specific antiviral activity.

## KHNYN$_{CTD}$ directly interacts with ZAP-X

Our data indicate that the N-terminal portion of KHNYN binds TRIM25, multiple KHNYN domains can interact with ZAP, and the NYN domain contains an endonucleolytic active site. While the CTD clearly contributes to antiviral activity and specificity (Fig. 3), its role in the formation of the antiviral complex is unclear. Previous reports suggest that the CTD binds NEDD8 and may function as a nuclear export signal[27,32]. We expressed and purified recombinant forms of ZAP-X and the KHNYN CTD and used size-exclusion chromatography (SEC) to test whether the CTD interacts with ZAP-X. Mixing CTD and ZAP-X resulted in a shift in the SEC elution profile relative to either protein alone, indicating a species of higher molecular mass than either CTD or ZAP-X (Fig. 5a). By SDS-PAGE analysis, both CTD and ZAP-X were present in these fractions, suggesting robust complex formation (Fig. 5a).

To determine the molecular details of this interaction, we solved a 2.30-Å crystal structure of ZAP-X bound to KHNYN$_{CTD}$ (Fig. 5b and Supplementary Table 2). Notably, the CTD interacts with a surface of ZAP-X on the opposite side from the binding site for CpG dinucleotides at ZnF2 and does not alter the ZAP-X structure[8]. Thus, KHNYN$_{CTD}$ binding would not interfere with CpG-specific RNA binding (Supplementary Fig. 5a). Moreover, ZAP-X interacts with a CTD surface opposite the linkage to the NYN domain. Thus, the presence of the NYN domain in NYN-CTD would not interfere with the ZAP-X binding site observed in the crystal structure. KHNYN$_{CTD}$ formed two hydrogen bonds with ZAP-X: a salt bridge between ZAP-X Lys12 and KHNYN$_{CTD}$

Asp695 (Fig. 5c, Interface 1) and another between the amide backbone of Ala31 in ZAP-X and the KHNYN$_{CTD}$ C-terminal carboxyl group of Phe719 (Fig. 5c, Interface 2). Additionally, a hydrophobic patch on KHNYN$_{CTD}$, including Phe696 and Phe719, binds in a hydrophobic groove on ZAP-X lined by residues Val6, Phe9, Ile30, and Leu32 (Fig. 5c, Interface 2). Phe655 of KHNYN$_{CTD}$ occupies a hydrophobic pocket of ZAP-X, packed against Gly128. Any amino acid larger than glycine at this position would prevent this packing. The ZAP-X/KHNYN$_{CTD}$ structure revealed a zinc-binding site formed by the two proteins, with the zinc atom coordinated by ZAP-X Glu29 and His17, KHNYN$_{CTD}$ His692, and a water molecule (Fig. 5c, Interface 3). The presence of zinc in this site was confirmed by zinc anomalous scattering. Our structure of KHNYN$_{CTD}$ bound to ZAP-X differs significantly from the published structure of the KHNYN$_{CTD}$ (PBD 2N5M), reported as a binding partner of NEDD8[27]. In our crystal structure, the KHNYN$_{CTD}$ has the traditional three-helix bundle structure of other CUE ubiquitin binding domains[33,34] but the previous KHNYN$_{CTD}$ structure has helices of different connectivity, spatial arrangement, and length (Supplementary Fig. 5f).

To probe the requirement of these interfaces for antiviral activity, we generated a panel of structure-based mutants and tested their activity in the ZAP$^{-/-}$ TRIM25$^{-/-}$ KHNYN$^{-/-}$ N4BP1$^{-/-}$ HEK293T cell reconstitution assay in the presence and absence of TRIM25. First, we tested the importance of ZAP amino acids at the ZAP/KHNYN interface. Three single amino acid substitutions, at Lys12, His17, and Gly128, reduced or eliminated TRIM25-independent antiviral activity (Fig. 5d). In the presence of TRIM25, Lys12, and His17 substitutions reduced HIV-1$_{CG}$ antiviral activity, (Fig. 5e). Similarly, amino acid substitutions in KHNYN at the ZAP binding interface, Phe696 and Phe719, eliminated TRIM25-independent activity (Supplementary Fig. 5b). However, none of the individual KHNYN substitutions tested affected antiviral activity in the presence of TRIM25 (Supplementary Fig. 5c). All variants tested were expressed at similar levels (Supplementary Fig. 5d, e). These results suggest that the KHNYN:ZAP interface observed in the crystal structure contributes to the antiviral activity of ZAP and KHNYN but is likely one of several redundant intermolecular interactions. This idea is supported by reconstitution assays showing that ZAP exhibited residual antiviral activity when coexpressed with ΔKH, ΔCTD, and NYN-CTD truncated variants of KHNYN (Fig. 3b). Furthermore, co-immunoprecipitation data showed that KHNYN$_{ΔCTD}$ can coprecipitate ZAP-L in the absence of TRIM25 (Fig. 3e). Together, these data suggest the crystal structure illuminates one interaction site between ZAP and KHNYN, but also that the assembly of a ZAP-KHNYN complex involves additional determinants.

## TRIM25 determinants that enhance ZAP antiviral activity

Like other TRIM proteins, TRIM25 encodes a central coiled-coil domain that drives the formation of obligate antiparallel dimers, and an N-terminal RING domain, which catalyzes autoubiquitination and ubiquitination of other target proteins (Fig. 6a)[25]. The RING domain is also an additional site of dimerization[24], and TRIM25, therefore, likely

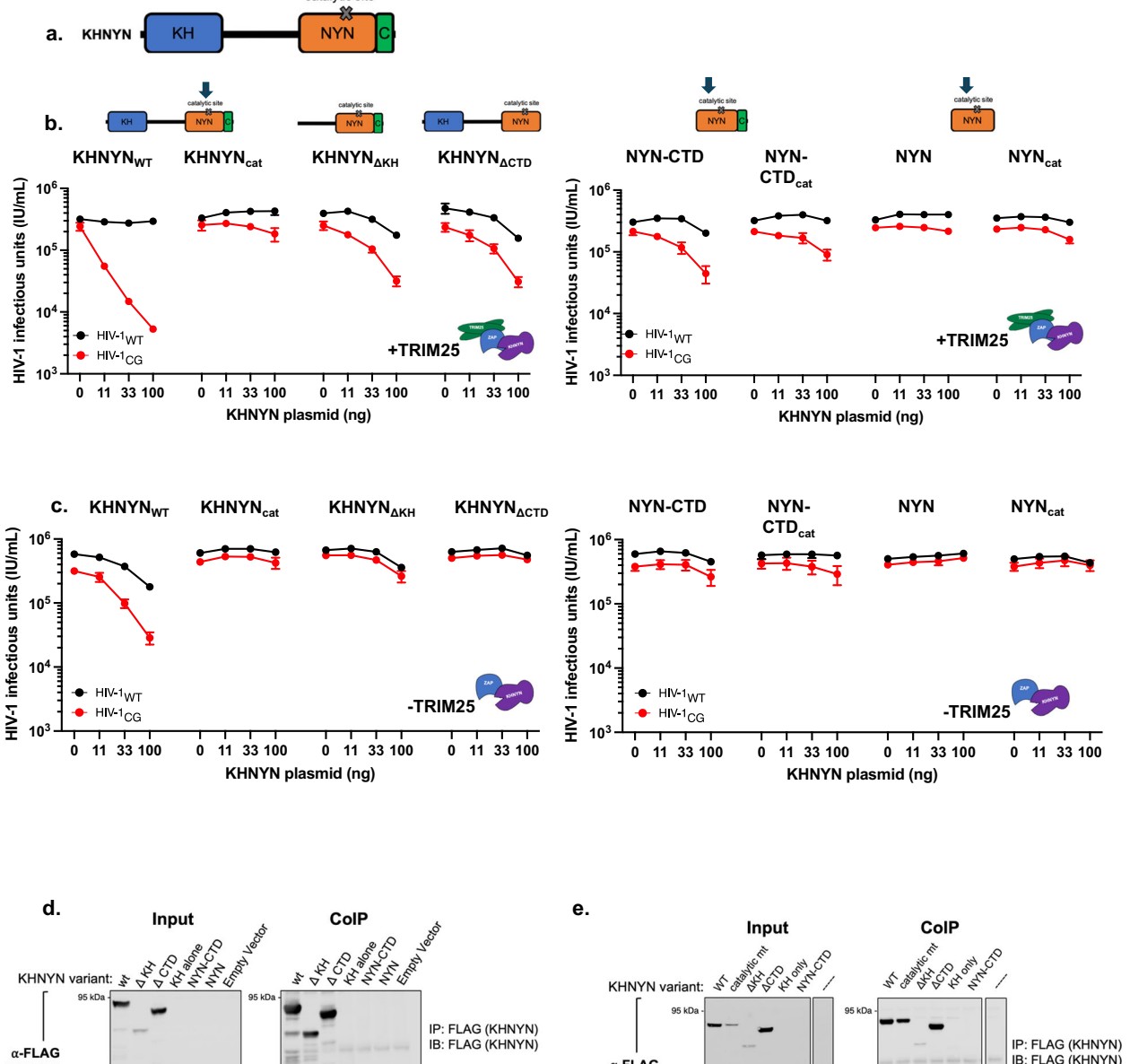

**Fig. 3 | KHNYN domain requirements for complex assembly and antiviral activity. a** Schematic showing KHNYN domain organization with an N-terminal KH domain (blue), a NYN domain (orange), and putative endonuclease catalytic site and a C-terminal domain (CTD, green) with homology to formerly annotated CUBAN domains. **b** Reconstitution experiments in which increasing amounts of KHNYN expression plasmids (ng) were co-transfected with constant amounts of TRIM25 (75 ng) and ZAP-L (150 ng) expression plasmids. Effects on HIV-1$_{WT}$ and HIV-1$_{CG}$ virus yield was measured as infectious units (IU/ml). Data were reported as mean ± sem of $n = 5$ biological replicates. **c** Same as (**b**) except that the TRIM25

expression plasmid was omitted. **d** Co-immunoprecipitation of TRIM25 with KHNYN-FLAG following transfection of 293T ZAP$^{-/-}$ TRIM25$^{-/-}$ KHNYN$^{-/-}$ N4BP1$^{-/-}$ cells with plasmids expressing the KHNYN-FLAG proteins described in (**a, b**). Proteins were immunoprecipitated from cell lysates with an anti-FLAG antibody and subjected to western blot analyses. **e** Co-immunoprecipitation of ZAP-L-3xHA with KHNYN-FLAG following transfection of 293T ZAP$^{-/-}$ TRIM25$^{-/-}$ KHNYN$^{-/-}$ N4BP1$^{-/-}$ cells with plasmids expressing the KHNYN-FLAG proteins described in (**a, b**). Proteins were immunoprecipitated from cell lysates with an anti-FLAG antibody and subjected to western blot analyses.

forms higher-order multimers. A C-terminal SPRY domain is a likely site of TRIM25-ZAP binding[16,26]. In a homologous protein, TRIM69, RING-driven multimerization is critical for antiviral activity[22] and the RING domain hydrophobic amino acids that drive TRIM69 multimerization are conserved in TRIM25 (Leu63, Leu69, Val72), suggesting a similar role in higher-order multimer formation (Fig. 6b). To delineate the

roles of RING domain multimerization and ubiquitination in ZAP-dependent antiviral activity, we generated a series of TRIM25 variants and tested their activity in the ZAP$^{-/-}$ TRIM25$^{-/-}$ KHNYN$^{-/-}$ N4BP1$^{-/-}$ HEK293T cell reconstitution assay. A TRIM25$_{\Delta RING}$ truncated protein did not enhance ZAP activity (Fig. 6c), suggesting that the RING domain is important for ZAP antiviral function. An X-ray crystal

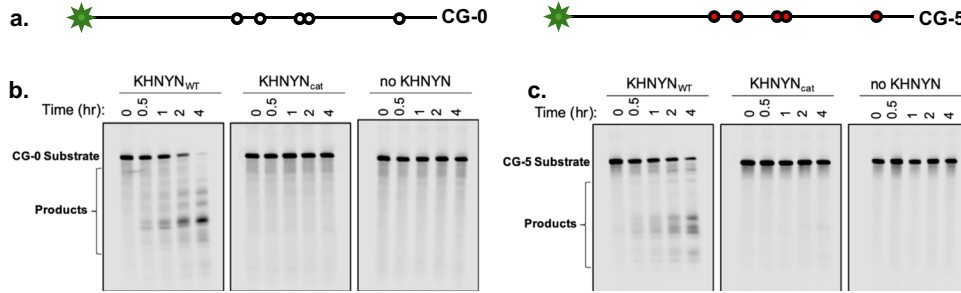

**Fig. 4 | KHNYN nuclease activity. a** Schematic representation of RNA substrates for KHNYN nuclease assays. Positions of CpG dinucleotides are in red on the CG-5 58-mer (right panel), with corresponding ApA substitutions in white on the CG-0 56-mer (left panel). **b**, **c** In vitro nuclease assay using KHNYN_WT-FLAG (**b**) and KHNYN_cat-FLAG (D565A/D566A) (**c**) purified by FLAG affinity chromatography from lysates of 293T ZAP$^{-/-}$ TRIM25$^{-/-}$ KHNYN$^{-/-}$ N4BP1$^{-/-}$ cells Nuclease activity was measured for each protein against the single-stranded CG-0 substrate over a 4-h time course. Activity showed vs a mock reaction with no protein added for comparison.

structure of the TRIM25 RING domain as a dimer in complex with Ubc13 and ubiquitin has been determined, and residues that are important for these interactions and ubiquitination activity identified[24]. Thus, we used structure/function guided mutagenesis to test which amino acids and activities are required for the TRIM25 RING domain to support ZAP antiviral activity. We first tested the effects of substituting hydrophobic amino acids at the TRIM25 RING-RING dimer interface (Leu63, Leu69, Val72)[24]. Most single and double amino acid substitutions at these sites resulted in diminished antiviral activity against HIV-1_CG, suggesting that TRIM25 RING-RING multimerization is important for the function of the ZAP antiviral complex (Fig. 6c). Conversely, substitutions at residues involved in ubiquitin binding (Asp10, Arg54, Lys65, and Asn71) and ligase activity (Cys50/53Ser)[24] or substitution of an acceptor site for autoubiquitination (Lys117)[35] had little or no effect on ZAP-dependent antiviral activity (Supplementary Fig. 6a). All variants were expressed to similar levels with the exception of the Lys117 mutant, suggesting that autoubiquitination, while not important for antiviral activity, is important for protein stability (Supplementary Fig. 6b, c). We tested two deletion mutants, TRIM25_ΔSPRY and TRIM25_ΔSPRY/7K, to elucidate the role of additional domains likely involved in ZAP binding (i.e., the SPRY domain) or possible RNA binding (seven Lys residues N-terminal to the SPRY domain)[16,36]. Both deletion mutants failed to enhance ZAP antiviral activity. Conversely, a mutant encoding an Ala substitution at each of the seven Lys residues[36] retained ZAP-enhancing activity (Fig. 6c and Supplementary Fig. 6d). These results confirm that the SPRY domain of TRIM25 is critical for ZAP-dependent antiviral activity and show that the TRIM25 residues previously reported to bind RNA are not important in this context.

To elucidate TRIM25 determinants required for association with ZAP and KHNYN assembly, we performed a series of co-immunoprecipitation experiments in which various TRIM25 constructs were expressed in ZAP$^{-/-}$ TRIM25$^{-/-}$ KHNYN$^{-/-}$ N4BP1$^{-/-}$ HEK293T-derived cells in which FLAG-tagged KHNYN or ZAP-L expression was dox-inducible. Truncated TRIM25-derived proteins containing the coiled-coil (CC) domain and either the B-box (BB) or SPRY domains were co-immunoprecipitated with FLAG-KHNYN, (Fig. 6d) or FLAG-ZAP-L (Fig. 6e). The BB-CC and ΔSPRY TRIM25 fragments could also be co-immunoprecipitated with ZAP-N (Supplementary Fig. 6e), albeit with reduced efficiency compared to ZAP-L (Fig. 6e). Together, these findings suggest that interactions between TRIM25 and other components of the ZAP antiviral complex are multivalent, and that simple TRIM25 binding interactions with ZAP or KHNYN are insufficient to explain its ability to stimulate antiviral activity.

Because the TRIM25 N-terminal RING and C-terminal SPRY domains were both essential for ZAP-enhancing activity, we next generated a panel of chimeric proteins in which either the RING or SPRY domains were exchanged with those of heterologous TRIM

proteins, specifically TRIM5, TRIM21, TRIM34, or TRIM38. Notably, all heterologous RING domains except TRIM34 supported potent and specific anti-HIV-1_CG antiviral activity in the context of these chimeras (Fig. 6f and Supplementary Fig. 6g). This finding suggests that a property common to TRIM protein RING domains rather than the identity of the RING itself, (likely RING-RING dimerization) is critical for antiviral activity. In contrast, none of the heterologous SPRY domains functionally substituted for the TRIM25 SPRY domain (Fig. 6g and Supplementary Fig. 6h), suggesting that the TRIM25 SPRY domain contains elements that confer specific interaction and enhance ZAP activity. None of the full-length TRIM proteins that served as a source of the RING- or SPRY- domains in the TRIM25-based chimeras exhibited TRIM25-like ability to enhance ZAP antiviral activity (Supplementary Fig. 6f).

## Reconstitution of antiviral activity with chimeric ZAP/TRIM25 or ZAP/KHNYN proteins

In principle, a minimal antiviral protein that targets CpG-rich RNA for depletion could consist simply of two protein domains that encode (i) specific CpG RNA binding activity and (ii) nuclease activity. The ZAP RBD is sufficient and essential for RNA binding activity (and antiviral activity, provided that both KHNYN and TRIM25 are present), and the NYN domain of KHNYN is sufficient and essential for in vitro nuclease activity. The remaining domains of ZAP and KHNYN, as well as TRIM25, may, therefore, simply provide protein domains that orchestrate the assembly of a complex that contains these two functional elements.

We, therefore, constructed a potential minimal CpG-targeting antiviral protein encoding the ZAP RBD and the nuclease domain of KHNYN (ZAP-X-KHNYN_NYN, Fig. 7a). This fusion protein exhibited potent antiviral activity in the ZAP$^{-/-}$ TRIM25$^{-/-}$ KHNYN$^{-/-}$ N4BP1$^{-/-}$ HEK293T cell reconstitution assay (Fig. 7b), and antiviral activity was dependent on the integrity of the NYN domain catalytic site. While this artificial antiviral protein was more potent against HIV-1_CG than HIV-1_WT, the HIV-1_CG specificity of the antiviral activity of ZAP-X–KHNYN_NYN was relaxed as compared to the complete reconstituted ZAP, TRIM25, and KHNYN combination (Figs. 1n, 7b, c). Unlike unfused ZAP-N or ZAP-L coexpressed with full-length KHNYN, the activity and specificity of the fused ZAP-X–KHNYN_NYN were unaffected by co-expression of TRIM25 (Fig. 7b). Thus, the activity-enhancing activity of TRIM25 can be bypassed by fusion of the active domains of ZAP and KHNYN, but an antiviral protein configured in this way does not fully recapitulate the specificity of the authentic ZAP, TRIM25, and KHNYN combination, further suggesting the importance of all three proteins in antiviral specificity.

The inclusion of TRIM25 caused both enhanced ZAP antiviral activity and enhanced specificity (Fig. 1) in a manner that was dependent on the TRIM25 SPRY domain (Fig. 6a, g). Additionally, the ZAP N-terminal RBD was essential for interaction with TRIM25 and both

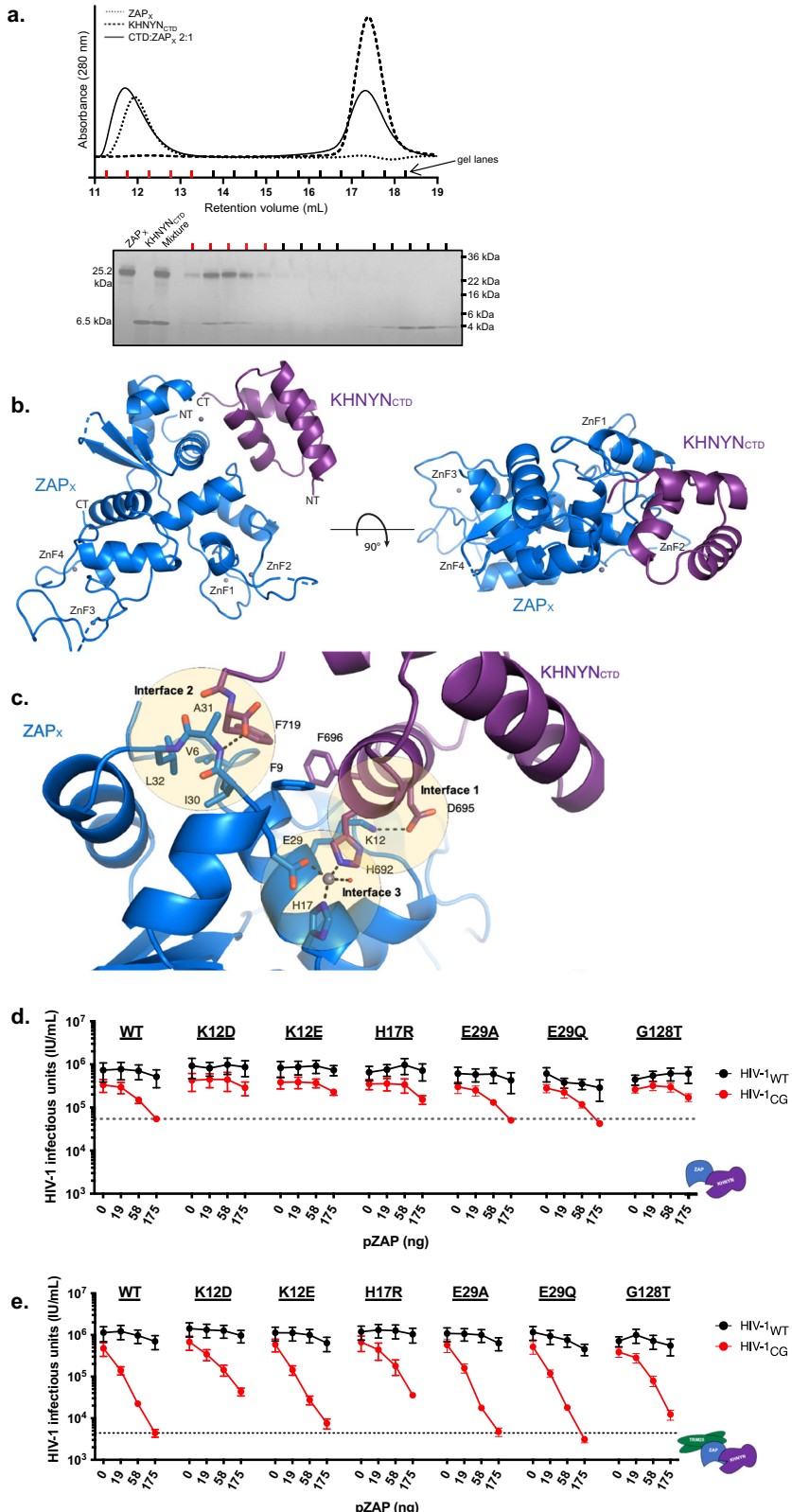

necessary and sufficient for interaction with KHNYN (Fig. 2d) and with RNA. If the function of TRIM25 is to facilitate the assembly of multimeric protein complexes containing ZAP and KHNYN, through self-multimerization, coupled with specific interactions with ZAP and KHNYN, then the TRIM25 SPRY domain might become dispensable if TRIM25-mediated complex assembly was instead driven by a TRIM25-based fusion protein in which the SPRY domain is replaced by the ZAP

N-terminal RBD (Fig. 7d) in such a context, an independent ZAP protein might also become dispensable. Therefore, we asked whether a TRIM25$_{\Delta SPRY}$–ZAP$_{RBD}$ fusion protein coexpressed with KHNYN could recapitulate antiviral activity and specificity of the ZAP, TRIM25 and KHNYN combination in the HEK293T ZAP$^{-/-}$ TRIM25$^{-/-}$ KHNYN$^{-/-}$ N4BP1$^{-/-}$ reconstitution assay. In fact, TRIM25$_{\Delta SPRY}$–ZAP$_{RBD}$ coexpressed with KHNYN yielded antiviral activity that was specific for

**Fig. 5 | Structure of ZAP_X−KHNYN_CTD complex. a** Formation of a ZAP_X-KHNYN_CTD complex. Size-exclusion chromatography profiles of ZAP_X (dotted trace), KHNYN_CTD (dashed trace) and a 1:2 mixture of ZAP_X and KHNYN_CTD (solid trace) illustrate the larger complex eluting in the protein mixture, consistent with the SDS-PAGE analysis of fractions from the ZAP_X-KHNYN_CTD mixture shown below the chromatography profile. Red lines mark the fractions containing the ZAP_X-KHNYN_CTD complex. One experiment was performed. **b** Overall structure of the ZAP_X-KHNYN_CTD complex. ZAP_X is shown in blue and KHNYN_CTD is shown in purple. Zinc ions are shown as gray spheres. **c** Details of the interaction between ZAP_X (blue) and KHNYN_CTD (purple) highlighting the three interacting interfaces. Hydrogen bonds are shown in dashed lines. The zinc (gray sphere) binding site formed by H17 and E29 on ZAP_X and H692 on KHNYN_CTD is shown with dashed lines. A water molecule bound to zinc is shown as a red sphere. **d** Reconstitution experiments in which increasing amounts of plasmids (ng) expressing WT and point mutant ZAP proteins informed by the crystal structure were co-transfected with a constant amount (75 ng) of KHNYN expression plasmid in 293T ZAP$^{-/-}$ TRIM25$^{-/-}$ KHNYN$^{-/-}$ N4BP1$^{-/-}$ cells. Effects on HIV-1_WT and HIV-1_CG virus yield was measured as infectious units (IU/ml). Data were reported as mean ± sem of $n = 5$ biological replicates, (**e**) Same as (**d**) except that 75 ng TRIM25 expression plasmid was also transfected.

HIV-1_CG, to a similar degree as that exhibited by co-expression of ZAP-N along with full-length TRIM25 and KHNYN (Fig. 7e, protein levels shown in Fig. 7f). While the bipartite TRIM25_ΔSPRY−ZAP_RBD/KHNYN reconstitution did not fully recapitulate the specificity of the complete tripartite ZAP-L/TRIM25/KHNYN reconstitution, this result shows that the TRIM25 SPRY domain indeed becomes dispensable for antiviral activity if the ZAP N-terminal RBD is fused in its place. Thus, the activity and specificity-enhancing properties of TRIM25 are, in large part, conferred by the N-terminal portion of the protein that drives homo-multimerization.

## Discussion

Defining the requirement for TRIM25, KHNYN, and N4BP1 proteins enabled us to generate a robust reconstitution system to functionally examine the requirements for ZAP antiviral activity directed at CpG-enriched viral genomes. KHNYN was previously reported to be required for ZAP activity in a context-dependent manner[18], while N4BP1 was a hit, along with ZAP, in a screen for antiviral proteins in which a probe vector included a CpG-rich reporter gene[37]. We found that KHNYN and N4BP1, which are both NYN domain-containing proteins, are redundant cofactors for ZAP-dependent inhibition of CpG-enriched HIV-1 and EV71 in at least some cell contexts. While TRIM25 was not absolutely required for antiviral activity against CpG-enriched derivatives of HIV-1 and EV71, its presence enhanced the antiviral activity of ZAP, as previously suggested[16]. Thus, the derivation of a cell line in which ZAP, TRIM25, KHNYN, and N4BP1 were all ablated enabled us to reconstitute antiviral activity with manipulated versions of each of these proteins and to determine the contribution of individual protein domains, and their activities and interactions, to activity against viruses with CpG-rich genomes.

As previously reported, we found that the isolated N-terminal ZAP RBD exhibited antiviral activity[3,16–18]. Antiviral activity of the isolated ZAP RBD remained dependent on TRIM25 and KHNYN co-expression and, consistent with this finding, the ZAP RBD was required for interaction with both TRIM25 and with KHNYN. Nevertheless, less severely truncated forms of ZAP were better able to homo-multimerize and co-immunoprecipitate TRIM25, and full-length ZAP-S and ZAP-L exhibited greater specificity for CpG-enriched vs wild-type HIV-1 than did the isolated ZAP RBD. Thus, while the ZAP RBD is necessary and sufficient for antiviral activity, full-length ZAP-L contains additional domains that optimize antiviral complex assembly and specificity.

Full-length KHNYN was required for full ZAP-dependent antiviral activity in the cell-based reconstitution assay. Indeed, multiple KHNYN domains, including the N-terminal KH domain and the CTD, interacted with ZAP and TRIM25. The integrity of the crystal contacts between the KHNYN CTD and the ZAP RBD increased the antiviral activity of ZAP/KHNYN in the absence of TRIM25 but were dispensable in the presence of TRIM25. These results suggest that the ZAP RBD−KHNYN_CTD interface revealed by the crystal structure is one of multiple interfaces between ZAP and KHNYN, and future structural studies will be required to reveal and characterize additional ZAP-KHNYN contacts.

The inclusion of TRIM25 in the reconstitution assay revealed that its SPRY domain was essential for its enhancing activity, and it could not be functionally replaced by SPRY domains from several other TRIM proteins. We previously showed that the TRIM25 SPRY domain governs species-dependent compatibility−chicken and human ZAP proteins exhibited increased activity when coexpressed with TRIM25 proteins encoding chicken and human TRIM25 SPRY determinants, respectively[26]. Together, these findings suggest that the TRIM25 SPRY domain contains determinants that mediate specific interactions with ZAP. While RNA binding appears important for the ability of TRIM25 to function as a RIG-I cofactor[36], TRIM25 RNA binding activity was dispensable in the context of ZAP antiviral activity. Additionally, TRIM25 is an active ubiquitin ligase, but amino acid residues that are crucial for interaction with the Ubc13 E2 enzyme and with ubiquitin, and are important for its ubiquitin ligase activity[24] were dispensable for its ability to enhance ZAP antiviral activity. Conversely, TRIM25 RING-RING contacts defined in crystal structures[21,24] were critical for the enhancement of antiviral activity. The TRIM25 RING domain could be functionally substituted with the RING domains of several TRIM protein, including TRIM69. Analogous RING-RING contacts in the TRIM69 RING domain drive high-order multimerization and the formation of extended TRIM69 filaments in cells[22]. The RING domain was not required for TRIM25 to interact with ZAP or with KHNYN. Together, these observations suggest that a major role for TRIM25 in ZAP anti-viral function is to drive protein complex multimerization. Other activities (RNA binding and ubiquitin ligase activity) that have previously been ascribed to TRIM25 and appear important for their role in function in other cellular activities[24,36] do not appear important for ZAP antiviral activity.

Protein−protein interactions can serve to physically link two activities residing in separate polypeptides into a single complex harboring both activities. In the case of the ZAP-containing complexes, a minimally active protein or complex should contain specific RNA recognition and destruction activities. As described here and elsewhere, CpG-specific RNA binding activity resides in the ZAP N-terminal RBD, while the NYN domain of KHNYN has RNA-specific nucleolytic activity. Linking a specific RNA binding domain to a nuclease domain could be envisaged as a general way to generate specific RNA-depleting proteins that are programmable based on the specificity of the RNA binding domain. Such proteins might be useful research tools or even therapeutic proteins.

Our attempts to generate a minimal CpG-specific antiviral protein by fusing the ZAP RBD to the NYN domain of KHNYN yielded a protein with antiviral activity but lacked the full CpG-rich RNA specificity achieved when full-length ZAP, KHYNYN, and TRIM25 were all present. These data are consistent with the idea that the complete ZAP, TRIM25, and KHNYN proteins not only link specific RNA binding domains to nuclease activity, but also suggest a model in which ZAP−ZAP interactions and TRIM25 driven multimerization functions to spatially organize multiple CpG binding sites present in multimerized ZAP proteins to enable multivalent interactions with a CpG-rich RNA target (Fig. 8). Our previous observations have shown that the isolated monomeric N-terminal ZAP RBD has a CpG binding pocket embedded within a larger RNA binding surface, and has a high affinity for an RNA oligonucleotide containing a single CpG dinucleotide, but retains

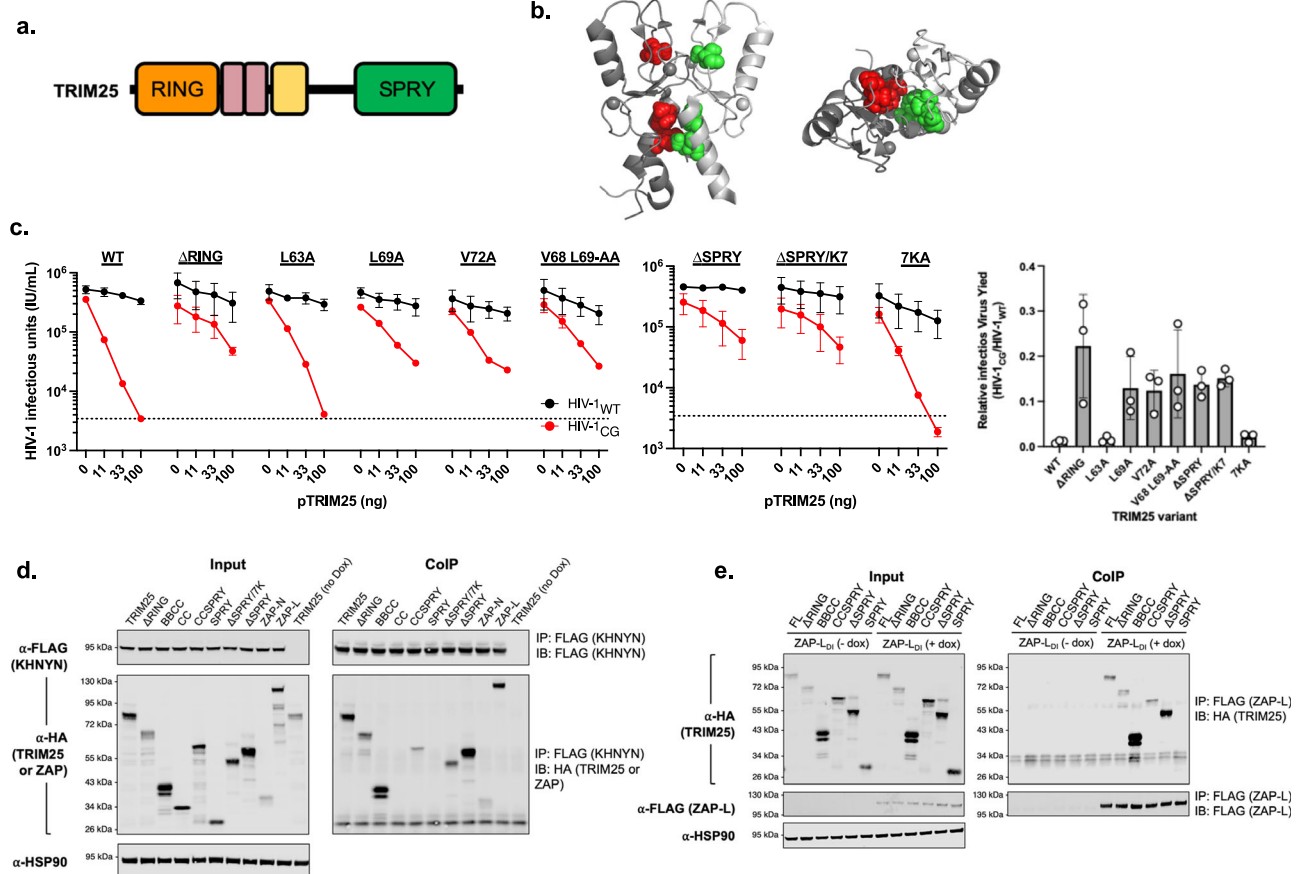

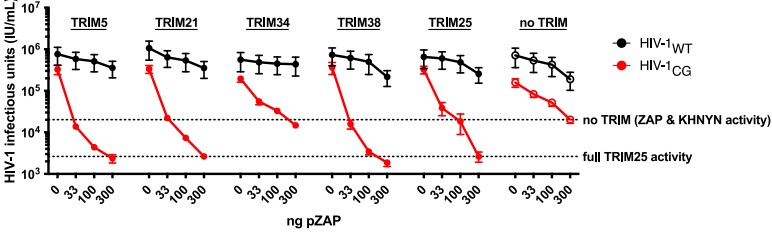

**f.  TRIM25 with RING domain exchanged from other TRIM proteins**

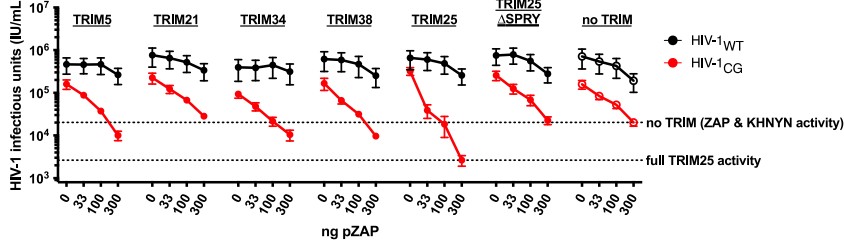

**g.  TRIM25 with SPRY domain exchanged from other TRIM proteins**

some residual (~10-fold reduced) affinity for an RNA oligonucleotide lacking CpG dinucleotides[8]. Moreover, our previous work has also shown that CpG numbers, spacing, and presence in an RNA context are all important for conferring sensitivity to ZAP, with ~15 CpG dinucleotides spaced ~12–30 nucleotides apart in a candidate RNA target that is primarily single-stranded representing an optimal target (Fig. 8)[30]. Again, this finding suggests that multivalent interactions between the ZAP-containing complexes and their RNA targets are important for CpG-specific antiviral function, and are enhanced by

ZAP–ZAP interactions and TRIM25 driven multimerization. It is also possible that the interaction between ZAP and its cofactors in the context of protein complexes provides a regulatory mechanism to restrain nuclease activity that is relieved by multivalent interaction between KHNYN and target CpG-rich RNAs. Future work, including the determination of the structure of a complex containing full-length ZAP, TRIM25, and or KHNYN, should further illuminate how these proteins recognize and eliminate CpG-rich viral RNA from infected cells.

**Fig. 6 | Molecular determinants of TRIM25 function. a** Schematic representation of TRIM25 highlighting the N-terminal RING domain (orange), two B-box domains (brown), a coiled-coil domain (yellow) that drives inherent dimerization, and a C-terminal SPRY domain (green). **b** Structure of the TRIM25 RING-domain dimer (from PDB 5EYA) with each subunit displayed in a different gray shade and hydrophobic amino acids at the dimer interface (Leu63, Leu69, and Val72) colored red and green on the respective subunits. **c** Reconstitution experiments in which increasing quantities of plasmids (ng) expressing wildtype and point mutant TRIM25 were co-transfected with a constant amount of plasmids expressing KHNYN (75 ng) and ZAP-L (150 ng). Wild-type TRIM25 was compared with a deletion of the RING domain, point mutants at hydrophobic residues in the RING domain (L63A, L69A, V72A, and V68A/L69A), a TRIM25 SPRY and proximal 7-Lys (7 K) deletion mutants were tested as well as a 7KA substitution mutant. Effects on HIV-$1_{WT}$ (black) and HIV-$1_{CG}$ (red) virus yield were measured as infectious units (IU/ml). Data were reported as mean ± sem of $n = 3$ biological replicates. The right panel shows a summary of the relative infectious HIV-$1_{CG}$ virus yield (compared to HIV-$1_{WT}$) when 100 ng of each variant TRIM25 expression plasmid was transfected. **d, e** Co-immunoprecipitation of TRIM25 with KHNYN-FLAG (**d**) or ZAP-L-FLAG (**e**) following transfection of 293T ZAP$^{-/-}$ TRIM25$^{-/-}$ KHNYN$^{-/-}$ N4BP1$^{-/-}$ cells (stably transduced with doxycycline-inducible KHNYN-FLAG (**d**) or ZAP-L-FLAG expression constructs (**e**)) with plasmids expressing TRIM25 domain deletion mutant proteins. Proteins were immunoprecipitated from cell lysates with an anti-FLAG antibody and subjected to western blot analyses. **f, g** Reconstitution experiments in which increasing amounts of plasmids (ng) expressing TRIM25 whose RING domains (**f**) or SPRY domains (**g**) were substituted with the RING domain from the indicated TRIM proteins were co-transfected with a constant amount of plasmids expressing KHNYN (75 ng) and ZAP-L (150 ng). Effects on HIV-$1_{WT}$ (black) and HIV-$1_{CG}$ (red) virus yield was measured as infectious units (IU/ml). Data were reported as mean ± sem of $n = 4$ biological replicates.

## Methods

### Cell lines
Cell lines HeLa (CCL-2), 293T (CRL-3216), RD (CCL-136), were obtained from the ATCC and grown in DMEM, supplemented with 10% fetal bovine serum. MT4 cells (ARP-120) were obtained from the NIH HIV Reagent Program Cell line identities were routinely confirmed by visual inspection under a light microscope and all cell lines were periodically tested for mycoplasma and retrovirus contamination.

Knockout cell lines of ZAP, TRIM25, KHNYN, and N4BP1 were generated using CRISPR-Cas9 as previously described[7]. Briefly, cells were generated by transduction with a lenti-CRISPR vector followed by selection in 5 µg ml$^{-1}$ blasticidin. Single-cell clones were derived by limiting dilution and maintained in the appropriate media with 5 µg ml$^{-1}$ blasticidin. HeLa ZAP$^{-/-}$ knockout cells were generated using the following guide RNA sequence: 5′-GGCCGGGATCACCC-GATCGGTGG-3′ and have been described previously[7]. HeLa TRIM25$^{-/-}$ knockout cells were generated using the following TRIM25-targeting guide RNA sequence: 5′-GAGCCGGTCACCACTCCGTG-3′. Cell lines that include a KHNYN knockout (HeLa KHNYN$^{-/-}$ and HeLa KHNYN$^{-/-}$/N4BP1$^{-/-}$ dKO) were generated using the following KHNYN-targeting guide RNA sequence: 5′− GCGGGCCCCCCAGGTAGGCA−3′. Cell lines that include a N4BP1 knockout (HeLa N4BP1$^{-/-}$ and HeLa KHNYN$^{-/-}$/N4BP1$^{-/-}$ dKO) were generated using the following N4BP1-targeting guide RNA sequence: 5′- AGAAAGAGAATGTTACCCCA-3′. The cell lines described above were used for HIV-1 and EV71 replication assays.

Human embryonic kidney (HEK) 293T, HEK293T ZAP$^{-/-}$/TRIM25$^{-/-}$ cells[16] were maintained in DMEM and supplemented with 10% fetal bovine serum. HEK293T KHNYN$^{-/-}$/N4BP1$^{-/-}$ (dKO) and HEK293T ZAP$^{-/-}$/TRIM25$^{-/-}$/KHNYN$^{-/-}$/N4BP1$^{-/-}$ cells were generated using CRISPR-Cas9 with sequential knockout of KHNYN and N4BP1 using the guide RNA sequences noted above. Frameshift mutations in KHNYN and N4BP1 were confirmed by PCR amplification and sequencing of the genomic locus. Western blot analysis was performed for N4BP1, but not for KHNYN, as no antibodies of sufficient quality were identified. HEK293T ZAP$^{-/-}$/TRIM25$^{-/-}$/KHNYN$^{-/-}$/N4BP1$^{-/-}$ cells carrying doxycycline-inducible variants of C-terminally HA-tagged ZAP-L or ZAP-N or C-terminally FLAG-tagged KHNYN were generated by stable transduction cells with LKO expression vectors encoding cDNAs encoding the respective protein with synonymous substitution to confer sgRNA resistance, followed by selection with puromycin. Reconstituted cells were used as a pool and were incubated with doxycycline only at the time of the experiment.

### Virus production
For single-cycle replication experiments, HIV-1 stocks (both HIV-$1_{WT}$ and HIV-$1_{CG}$) were generated by transfection of 293T cells in 15-cm dishes with 15 µg of proviral plasmid and 1.5 µg of VSV-G expression plasmid using polyethyleneimine (Polysciences). HIV-$1_{WT}$ and HIV-$1_{CG}$ contained a GFP reporter as these were generated from HIV-$1_{NHG}$-derived proviral plasmids (accession number JQ585717). At 18–24 h post-transfection, cell culture media was replaced, and at 48-h post-transfection, supernatants were collected and filtered through a 0.22-µm filter. All virus stocks were stored at −80 °C and were thawed once at the time of use. Titers of viral stocks were determined by performing threefold serial dilutions and subsequent infections of MT4 cells in a 96-well plate ($5 × 10^4$ cells seeded per well). After 48 h, cells were fixed with 4% paraformaldehyde, and the percentage of GFP-positive cells was determined using flow cytometry and calculated using FlowJo.

EV-A71 viral stocks were generated as previously described[30,38] Briefly, viral plasmids were linearized with *MluI* restriction enzyme and used to generate viral RNA using the T7 RiboMAX Express large-scale RNA production system. Viral RNA was then transfected in ZAP-deficient RD cells using the TransIT-mRNA transfection kit. When the cytopathic effect was observed in ~80% of cells, supernatants were collected and filtered. EV-A71/A+ (EV71$_{WT}$) and EV-A71/CG-48/A+ (EV71$_{CG}$) NanoLuc reporter-containing viruses were selected, and working stocks were made by passaging previously generated stocks single time in ZAP-deficient RD cells[30]. Supernatants from passages were collected and filtered through a 0.1-µm filter after ~80% of cells were observed to be cytopathic. All virus stocks were stored at −80 °C and were thawed once at the time of use. Viruses were titered onto each HeLa cell line of interest by performing fivefold serial dilutions and subsequent infections in a 96-well plate ($5 × 10^4$ cells seeded per well). Four replicates of each serial dilution were performed.

### HIV-1 replication assays
For single-cycle infection experiments, Hela cells were plated in 96-well plates. of HIV-$1_{WT}$ and HIV-$1_{CG}$ viral stocks were threefold serially diluted, and 100 µL of each virus dilution was applied 18–24 h after plating cells. The next day, the cell media was aspirated, and cells were washed three times with 1xPBS before fresh media was replaced. At 48 h after infection, 100 µL of supernatant from each infection was transferred onto MT4-GFP indicator cells ($5 × 10^4$ cells per well) which were incubated for another 48 h. The initial infected Hela cell populations were washed in 1xPBS and fixed in 4% PFA. The percentage of GFP-positive cells was assessed by flow cytometry and quantified using FlowJo to determine the multiplicity of infection (MOI) on a given Hela-derived cell line in each experiment. At 48 h after infection, MT4-GFP cells were also fixed in 4% PFA, analyzed by flow cytometry, and quantified using FlowJo to determine the infectious units produced by a given HIV-$1_{WT}$ or HIV-$1_{CG}$ infected Hela cell line. Experiments were performed with technical duplicates and were performed at least three times.

### EV71 replication assays
HeLa cells and knockouts thereof were plated in a 12-well plate at $1.5 × 10^5$ cells per well and were infected with either EV71$_{WT}$ or EV71$_{CG}$ at

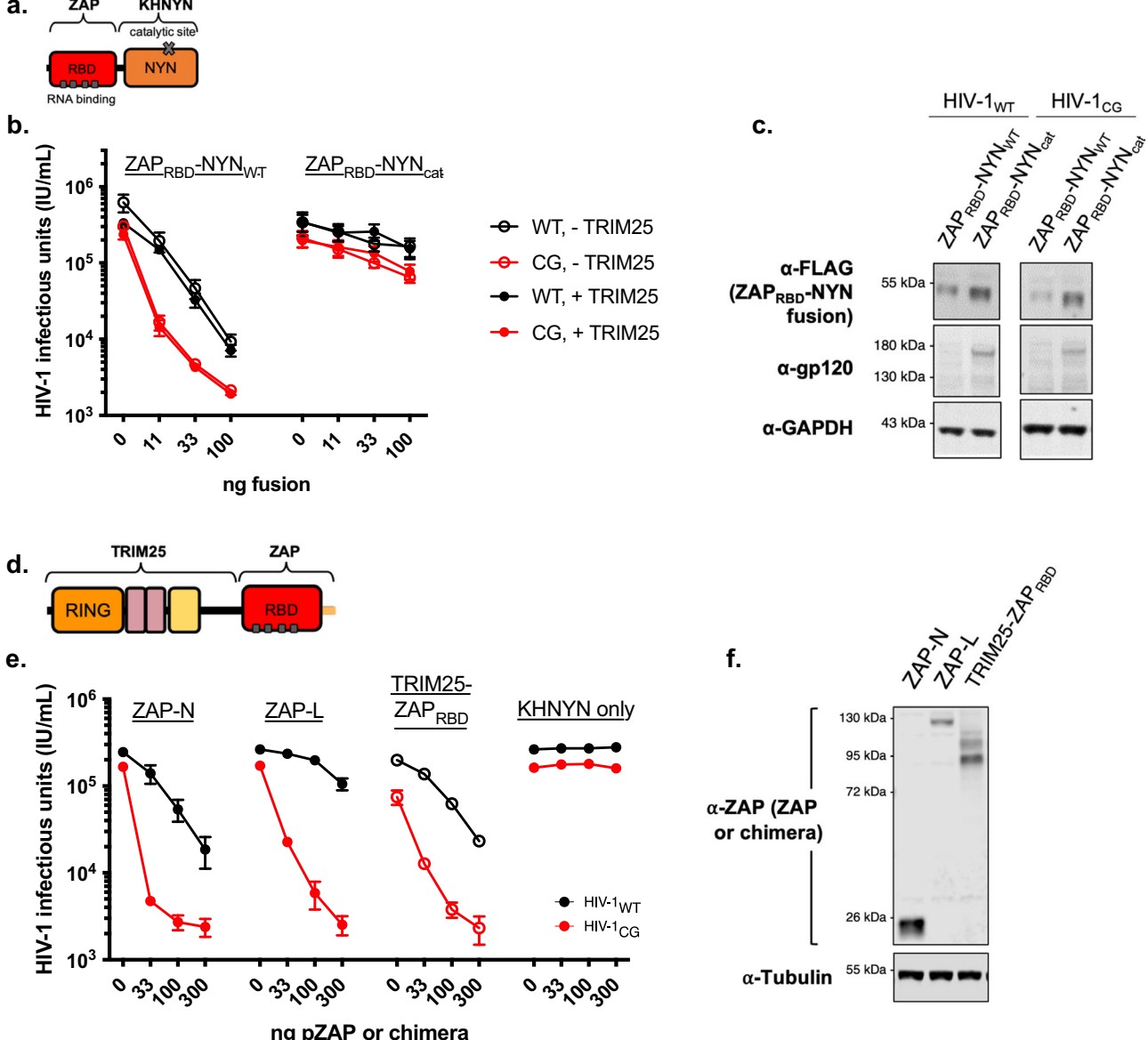

**Fig. 7 | Reconstitution of antiviral activity with designed chimeric antiviral proteins. a** Schematic representation of a ZAP-KHNYN chimera constructed by fusing the ZAP RNA binding domain (RBD) to the catalytic NYN domain from KHNYN. **b** Reconstitution experiments in which 293T ZAP⁻/⁻ TRIM25⁻/⁻ KHNYN⁻/⁻ N4BP1⁻/⁻ cells were transfected with increasing amounts of plasmids (ng) expressing the ZAP$_{RBD}$-NYN chimera in the absence of KHNYN or ZAP, in the presence or absence of a TRIM25 expression plasmid (75 ng). Effects on HIV-1$_{WT}$ (black) and HIV-1$_{CG}$ (red) virus yield was measured as infectious units (IU/ml). Data were reported as mean ± sem of $n$ = 4 biological replicates. **c** Lysates from 293T ZAP⁻/⁻ TRIM25⁻/⁻ KHNYN⁻/⁻ N4BP1⁻/⁻ cells, transfected as in Fig. 7b were analyzed by western blotting probing with antibodies against RBD-NYN-FLAG, HIV-1 gp120, and

GAPDH. **d** Schematic representation of a TRIM25/ZAP chimera constructed by fusing the N-terminus of TRIM25 (RING-B-box-Coiled-Coil) with the ZAP RBD. **e** Reconstitution experiments in which 293T ZAP⁻/⁻ TRIM25⁻/⁻ KHNYN⁻/⁻ N4BP1⁻/⁻ cells were transfected with increasing amounts of plasmids (ng) expressing the TRIM25-ZAP$_{RBD}$ chimera plus KHNYN in the absence of ZAP or TRIM25. For comparison, cells were transfected with ZAP-N and ZAP-L, plus TRIM25 and KHNYN, or KHNYN alone. Effects on HIV-1$_{WT}$ (black) and HIV-1$_{CG}$ (red) virus yield was measured as infectious units (IU/ml). Data were reported as mean ± sem of $n$ = 4 biological replicates. **f** Lysates from 293T ZAP⁻/⁻ TRIM25⁻/⁻ KHNYN⁻/⁻ N4BP1⁻/⁻ cells, transfected as in Fig. 7e were analyzed by western blotting probing with antibodies against TRIM25-RBD-3xHA and Tubulin.

an MOI of 0.02 for 1 h at 37 °C. Cells were then washed twice with 1xPBS and 1.5 mL of complete DMEM was replaced per well. Time points were collected every 12 for 72 h in total; at each timepoint, 100 μL of supernatant was collected and incubated with 25 μL of 5x concentrated passive lysis buffer (Promega). NanoLuc luciferase activity was measured using the Nano-Glo luciferase system (Promega), and relative light units were used as an indicator of the amount of luciferase released into the supernatant at a given timepoint. Experiments were performed in technical duplicate and were performed at least three times.

## ZAP, TRIM25, and KHNYN antiviral function reconstitution assays

To assess the function of various components of ZAP antiviral complexes, HEK293T ZAP⁻/⁻/TRIM25⁻/⁻/KHNYN⁻/⁻/N4BP1⁻/⁻ cells were seeded at 1.5 × 10⁵ cells per well in a 24-well-plate and transfected with 350 ng of HIV-1$_{WT}$ or HIV-1$_{CG}$ and a combination of the following components. When ZAP proteins were evaluated, varying amounts of ZAP expression plasmid were cotransfected with 75 ng of a KHNYN expression plasmid and 75 ng of a TRIM25 expression plasmid, unless otherwise specified. When KHNYN proteins were evaluated, varying

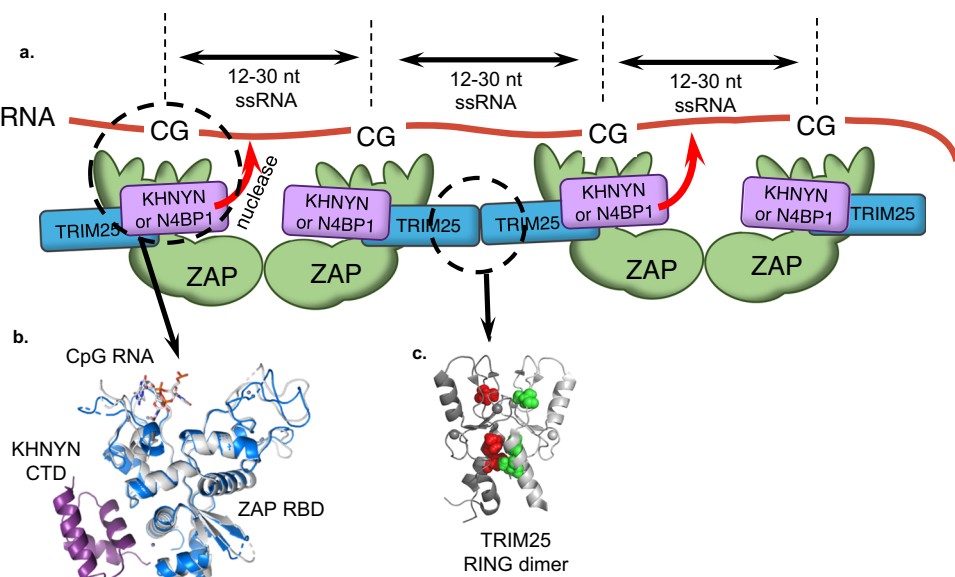

**Fig. 8 | Model for mechanism of action of ZAP, KHNYN, and TRIM25 in CpG-specific antiviral activity. a** Model for the assembly of ZAP, KHNYN (or N4BP1), TRIM25 and a target RNA that enables the nuclease to specifically deplete CpG RNAs that are recognized through multivalent interaction with ZAP-containing complexes whose multimerization is driven by ZAP:ZAP, ZAP:TRIM25, and TRIM25:TRIM25 interactions. **b** Details of the interactions between the core functional ZAP NTD a CpG dinucleotide containing RNA and the KHNYN CTD that enable monovalent recruitment of a KHNYN nuclease to a target RNA, (**c**) Details of the TRIM25 RING-RING interaction that contribute to the formation of ZAP containing complexes that recognize CpG-rich RNA in a multivalent manner.

amounts of KHNYN expression plasmid were cotransfected with 150 ng of a ZAP expression plasmid and 75 ng of a TRIM25 expression plasmid, unless otherwise specified. When TRIM25 proteins were evaluated, varying amounts of TRIM25 expression plasmid were cotransfected with 150 ng of a ZAP expression plasmid and 75 ng of KHNYN expression plasmid. At 24 h after transfection, media was replaced, and supernatants were collected and filtered using a 0.22-μm filter plate (Corning) at 48 h post-transfection. Infectious virus yield was determined using MT4-GFP indicator cells. Transfected cells were then lysed in NuPAGE buffer and protein samples were analyzed by western blotting.

## Western blot analyses

Cell lysates suspended in NuPAGE buffer were sonicated three times for 10 s each and then were incubated at 72 °C for 10 min. Protein samples were resolved on a 4–12% PAGE gel (Novex) using 1xMOPS running buffer. Protein was then transferred to a nitrocellulose membrane, blocked at room temperature in LiCor Intercept blocking buffer and then incubated with the following antibodies overnight at 4 °C: anti-Flag (mouse, 1:5000, Sigma F1804), anti-HA (rabbit, 1:5000, Rockland 600-401-384), anti-HSP90 (rabbit, 1:10,000, Proteintech 13171-1-AP), anti-Tubulin (mouse, 1:10,000, Sigma T5168), anti-HIV-1 gp120 (goat, 1:1000, American Research Products 12-6205-1), anti-ZC3HAV1 (rabbit, 1:2000, Proteintech 16820-1-AP), or anti-TRIM25 (rabbit, 1:5000, Bethyl A301-857A). Blots were washed three times with 1xPBS containing tween-20 (PBS-T) and then were incubated with secondary antibodies: anti-Mouse IgG IR700 or IR800 Dye Conjugated (LiCor), Anti-Rabbit IgG IR800 or IR700 Dye conjugated (LiCor), or Anti-Goat IgG horseradish peroxidase-conjugated (Jackson). Blots were washed another three times with 1xPBS-T and were imaged on the Licor Odyssey scanner (IR secondary antibodies) or were incubated with chemiluminescent horseradish peroxidase substrate and imaged on a CDigit blot scanner.

## Co-immunoprecipitation

To study the interactions between proteins of the ZAP antiviral complex, cells (either HEK293T or HEK293T ZAP$^{-/-}$/TRIM25$^{-/-}$/KHNYN$^{-/-}$/N4BP1$^{-/-}$) were seeded into a six-well dish at $1 \times 10^6$ cells per well and 24 h later, plasmids expressing the proteins of interest were transfected. Specifically, 3 μg ZAP expression plasmid, 2 μg KHNYN expression plasmid, and 1 μg TRIM25- expression plasmid were used as noted in a given experiment). In some experiments, in which stably transduced cells were used, expression of ZAP-L-3xHA, ZAP-N-3xHA, or KHNYN-FLAG was induced with Doxycycline (1:50,000) at the time of plating and transfections of plasmids containing other components of the complex were performed 24 h later. Cells were lysed 48 h after transfection in 500 μL Lysis Buffer (25 mM HEPES pH 7.0, 30 mM KCl, 2 mM EDTA, 0.5% Igepal CA-630 supplemented with protease inhibitor cocktails) and were treated with RNase A (Roche, 100 U) while cells lysed at 4 °C, gently rocking for 15 min. Protein G agarose beads were freshly prepared by pre-adsorption to either anti-HA (BioLegend 901501) or anti-FLAG (Sigma F1804) antibodies while cells were lysed. Pre-adsorbed beads were incubated with clarified lysates and incubated at 4 °C, gently shaking overnight. Magnetic beads were captured and washed twice in lysis buffer and three times in IP wash buffer (25 mM HEPES pH 7.0, 500 mM KCl, 2 mM EDTA, 0.5% Igepal CA-630). Protein complexes were resuspended in NuPAGE buffer and resolved on a 4–12% PAGE gel (Novex), transferred to nitrocellulose membranes and analyzed by western blotting as described above.

## Plasmid constructs

Full-length forms of ZAP, TRIM25, and KHNYN were cloned into the expression plasmid pCR3.1. All plasmids that were utilized for transient transfection experiments were pCR3.1-derived. Various mutants were generated using site-directed mutagenesis, while minimal domains, domain deletions, or fusion constructs derived from each of these proteins were generated by Gibson assembly of fragments of interest into pCR3.1, which had been linearized by restriction digest. All constructs were either tagged with three HA epitopes or one FLAG epitope, as indicated in a given experiment. Plasmids used for stable transduction and doxycycline-induced expression were cloned into the LKO lentivirus expression vector, and Gibson assembly was used to insert gene fragments of interest into LKO after restriction enzyme linearization as described with pCR3.1. A list of all recombinant plasmids used in this study and accompanying details are contained in Supplementary Table 1. Sequences of oligonucleotides used in molecular construction are contained in a Supplementary Data 1 file.

## Recombinant protein production and purification from mammalian cells

Six 15-cm dishes, each containing $8 \times 10^6$ HEK293T KHNYN$^{-/-}$/N4BP1$^{-/-}$ (dKO) cells, were transfected with 12 µg of pCR3.1-based plasmids expressing either KHNYN$_{WT}$-FLAG or KHNYN$_{cat}$-FLAG. Cells were harvested 48 h post-transfection by resuspension with 1xPBS and were then pelleted and stored at −80 °C. Cell pellets were resuspended in CelLytic M (Sigma) buffer supplemented with 200 mM NaCl, 2 mM DTT, and protease inhibitor cocktails. Resuspended cell lysates were rotated end-over-end for 1 h at 4 °C, and then lysates were cleared by centrifugation at $21,000 \times g$ for 1 h. KHNYN$_{WT}$-FLAG and KHNYN$_{cat}$-FLAG proteins were purified from cleared lysates using FLAG magnetic resin. Lysates were batch-bound to FLAG resin, rotating end-over-end overnight at 4 °C. FLAG resin was washed (1) twice with 20 mM HEPES pH 7 + 200 mM NaCl, (2) twice with 20 mM HEPES pH 7 + 1 M NaCl, (3) twice with 20 mM HEPES pH 7 + 200 mM NaCl + 0.1% Triton X-100) twice with 20 mM HEPES pH 7 + 200 mM NaCl. Purified protein was eluted with FLAG elution buffer (20 mM HEPES pH 7, 200 mM NaCl, 10% glycerol, and 1x FLAG peptide (Sigma). The purity of eluted protein was assessed by SDS-PAGE and quantified as compared to BSA standards on the same gel using ImageJ to generate a BSA standard curve. Eluted fractions were stored at −80 °C.

## Recombinant protein production and purification from *Escherichia coli* cells

DNA encoding the KHNYN$_{CTD}$ (residues 667–719) domain of KHNYN was cloned into a bacterial pMCSG7 expression vector encoding an N-terminal tobacco etch virus (TEV) protease cleavable His$_6$ tag, creating a p7KHNYN$_{CTD}$ plasmid. *E. coli* Rosetta2 cells (Novagen) were transformed with pKHNYN$_{CTD}$. Cultures were grown in TB media with 100 µg/mL ampicillin at 37 °C to an OD of 1.0, then cooled to 20 °C over 1 h. Expression was induced by the addition of 0.4 mM IPTG, and cultures were grown overnight at 20 °C. Cells were lysed in a buffer containing 25 mM Tris pH 8.0 and 200 mM NaCl by sonication. The protein was purified from the cleared cell lysate using Ni-NTA resin (Qiagen), and the N-terminal His$_6$ tag was cleaved with TEV protease. The protease reaction mixture was then run on Ni-NTA, followed by gel filtration on a Superdex 75 (16/60) column (GE Healthcare Life Sciences) in CTD storage buffer (20 mM HEPES pH 7.5, 200 mM NaCl, 5% glycerol, 1 mM tris(2-carboxyethyl)phosphine). Purified KHNYN$_{CTD}$ was flash-frozen in liquid nitrogen and stored at −80 °C. ZAP-X (residues 2-227) was purified as previously described[8]. Briefly, *E. coli* Rosetta (DE3) cells (Novagen) transformed the pMCSG7 expression vector encoding an N-terminal TEV-protease cleavable His6 tag, and the ZAP RBD and grown overnight with the addition of 50 µM zinc acetate. Cells were lysed by sonication in PBS buffer 0.5% CHAPS, 0.1% β-mercaptoethanol, 2 mM MgCl$_2$, benzonase, and protease inhibitors His6-TEV-hZAP RDB was purified from the using Ni-NTA resin (Qiagen) and the His6-TEV tag was cleaved with TEV protease. The cleavage reaction mixture was passed through a 5 mL HiTrap S HP column (GE Healthcare Life Sciences) followed by gel filtration on a Superdex 75 16/60 column (GE Healthcare Life Sciences) in a final buffer containing as above. The peak fractions containing ZAP RBD were flash-frozen in liquid nitrogen.

To visualize binding between ZAP-X and KHNYN$_{CTD}$, ZAP-X was incubated with KHNYN$_{CTD}$ at a 1:2 molar ratio for 1 h on ice and injected on a Superdex 75 (30/100) column (GE Healthcare Life Sciences) equilibrated with CTD storage buffer and run at 0.3 mL/min flow rate. As controls, ZAP-X or KHNYN$_{CTD}$ were separately injected onto the same Superdex 75 (30/100) column (GE Healthcare Life Sciences) equilibrated with the CTD storage buffer and run at 0.3 mL/min flow rate. Fractions eluted from the size-exclusion column analyzing ZAP-X:KHNYN$_{CTD}$ binding were also visualized by a Coomassie-stained SDS-PAGE gel.

## Crystallization and structure determination of ZAP-X−KHNYN$_{CTD}$ complex

ZAP-X was incubated with a 1.5-fold molar excess of KHNYN$_{CTD}$ for 1 h at 4 °C, concentrated, and run on an analytical Superdex 75 column in CTD storage buffer. The peak containing the complex was pooled, concentrated to 3.5 mg/mL and screened for crystallization conditions with an Art Robbins Gryphon Crystallization Robot. Crystals grew in 20% PEG 3350, 0.2 M NaCl, were cryoprotected in well solution with 30% ethylene glycol, and flash cooled in liquid nitrogen.

Diffraction data were collected at the Life Sciences Collaborative Access Team (LS-CAT) beamline 21-ID-G at the Advanced Photon Source, Argonne National Laboratory, and processed with HKL2000[39]. ZAP-X−KHNYN$_{CTD}$ crystallized in space group $P2_12_12_1$ with one complex in the asymmetric unit. The structure was solved by molecular replacement using Phaser[40] with the structure of ZAP-X (PDB 6UEI)[8] as a search model. Iterative rounds of molecular replacement failed to place the previously solved NMR structure of KHNYN$_{CTD}$ (PDB 2N5M)[27,28] in the structure. The KHNYN$_{CTD}$ structure was built manually into the electron density map using Coot[41], and the structure was completed with iterative rounds of automated refinement in Buster[42] and model building in Coot. Data statistics and model quality are summarized in Supplementary Table 2. The metal binding site site was confirmed to contain zinc in an anomalous difference map using data were collected at LS-CAT beamline 21-ID-D at X-ray energies above and below the zinc K absorption edge. The structure was validated with MolProbity[43].

## In vitro nuclease assay

Single-stranded oligonucleotides were purchased from Integrated DNA Technologies, 5′-FAM-labeled. In vitro nuclease assays of purified KHNYNwt-FLAG or KHNYNcat-FLAG against single-stranded RNA substrates were performed in buffer containing 25 mM Tris-HCl pH 7.5, 150 mM NaCl, 5% glycerol, 2.5 mM MgCl$_2$, 25 µM ZnCl$_2$, 400 nM KHNYNwt-FLAG, or KHNYNcat-FLAG and 1 µM RNA substrates. Reaction mixtures were incubated at 37 °C for 0, 30, 60, 120, and 240 min and quenched by the addition of a loading dye to a final composition of 42.5% formamide, 0.25 mM EDTA, 0.0125% (w/v) xylene cyanol, and 0.0125% (w/v) bromophenol blue. Samples were loaded directly onto a Novex 10% TBE-urea gel (Invitrogen) and resolved by denaturing gel electrophoresis in TBE (Tris/boric acid/EDTA) buffer. Fluorescence signals were acquired using a LiCor Odyssey in the 800 nm channel.

## Reporting summary

Further information on research design is available in the Nature Portfolio Reporting Summary linked to this article.

## Data availability

The Crystallography data generated in this study have been deposited in the Protein Data Bank database under accession code 9BGL. Previously published structures used in this manuscript are available in the Protein Data Bank under accession codes 6UEI, 2N5M, and 5EYA. The numerical data generated in this study are provided in the Supplementary Information and Source Data files. Oligonucleotide sequences are provided in Supplementary Data files. Source data are provided with this paper.

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

## Acknowledgements

This work was supported by NIH grant U54 AI170660 (to M.D.O., J.L.S., and P.D.B.). J.B. was supported by an NIH fellowship (F32AI160904). Z.Y. was supported by NIH T32 GM132046. P.D.B. is an HHMI Investigator. This research used resources from the Advanced Photon Source, a US Department of Energy (DOE) Office of Science User Facility operated for the DOE Office of Science by Argonne National Laboratory under Contract No. DE-AC02-06CH11357. The use of LS-CAT Sector 21 was supported by the Michigan Economic Development Corporation and the Michigan Technology Tri-Corridor (Grant 085P1000817). This article is subject to HHMI's Open Access to Publications policy. HHMI laboratory heads have previously granted a non-exclusive CC BY 4.0 license to the public and a sublicensable license to HHMI in their research articles. Pursuant to those licenses, the author-accepted manuscript of this article can be made freely available under a CC BY 4.0 license immediately upon publication.

## Author contributions

Conceptualization J.A.B., M.O., J.L.S., and P.B. Investigation: J.A.B., J.L.M., M.A., D.G.-C., and Z.C.Y., Writing—original draft: J.A.B., M.O., J.L.S., and P.B., Writing—review and editing: J.A.B., Z.C.Y., M.O., J.L.S., and P.B.

## Competing interests

The authors declare no competing interests.
