## [Peer Review File · Nature Communications]

REVIEWER COMMENTS

Reviewer #1 (Remarks to the Author):

In this study, the authors investigated the functional and physical interactions of ZAP with KHNYN and TRIM25, two ZAP co-factors that have been previously reported to be important for the antiviral activity of ZAP with the underlying mechanisms being not very clear. They find that multiple domains of KHNYN contribute to its interaction with ZAP. Furthermore, they provide experimental evidence showing that KHNYN is an active RNA nuclease. They also map the TRIM25 determinants that enhanced ZAP activity. While these findings represent an important advance in understanding how the antiviral activity of ZAP is executed and regulated, two major concerns need to be addressed. First, the authors use HIV-1WT as a negative control in the studies of the antiviral activity of ZAP. However, multiple results in this manuscript showed that HIV-1WT was inhibited by ZAP although the inhibitory effect was not as dramatic as on HIV-1CG. A likely explanation of these results is that the sensitivity of HIV-1WT to ZAP is just moderate and enhanced by the introduction of CG dinucleotide in HIV-1CG. This notion needs to be clarified in the manuscript to avoid confusion. In addition, both HIV-1CG and EV71CG used in this study are artificial viruses. To validate the biological relevance of the main findings, the authors need to show that KHNYN and TRIM25 are important for the antiviral activity of ZAP against a naturally existing virus, say Sindbis virus. This would tell us whether the observations are specific to the artificial viruses. Second, the authors repeatedly used the term ZAP/KHNYN/TRIM25 complex. They need to provide experimental results to show the existence of the complex, say by co-IP assays. In summary, the findings reported in this manuscript are potentially important but some issues need to be addressed to support the main conclusions.

Specific comments:

1. Figure 1, as said above, the authors need to use a naturally existing virus to test the requirement of TRIM25 and KHNYN for the antiviral activity of ZAP in these cells. The authors could compare the replication kinetics of the virus in the different KO cell lines. This should be an easy experiment.
2. In Figure 1m, HIV-1WT was also inhibited by about 10-fold. However, in Figure 1n, this inhibition was not observed, why?
3. Figure 7b, the ZAP(RBD)-NYN fusion protein displayed robust antiviral activity against HIV-1WT, why? This needs to be better explained.
4. Discussion: as said above, the authors need to show the existence of the complex.

Minor:

1. In Figure 1a-h, HIV-1WT seems to replicate better in HeLa-Ctrl cells than in the other cells, especially the ZAP KO cells, any explanation?
2. The Input and CoIP blots are not aligned.
3. There are many grammar errors that need to be fixed. For example: in the Abstract, “we could designed artificial.....”

Reviewer #2 (Remarks to the Author):

The zinc finger antiviral protein (ZAP) is a well-known and broad acting restriction factor, that targets CpG islands in virus-derived RNA for degradation. To this end, ZAP recognizes the target RNA then recruits co-factors like the nuclease KYHNYN or N4BP1. In addition, the E3 ligase TRIM25 is recruited to promote the function of the complex. However, how the subunits of this complex are coordinated was for far unknown.

In this manuscript, Bohn and colleagues systematically assesses the domains/functions required for the proper assembly of the complex as well as that both recruited nucleases work in a redundant manner. Furthermore, the functional interface of the complex is presented as a crystal structure. In addition, the authors construct a minimal functional complex containing of fused protein domains.

Overall this is a well-written and systematic study which provides novel insight into a fascinating antiviral factor complex. The data largely supports the conclusions of the manuscript. I have just a few major suggestions as well as a few minor comments to improve the current manuscript and clarify a few inconsistencies.

The big question is still what TRIM25 is doing in that complex to enhance its function since, ubiquitination seems to be not required and RNA binding function is dispensable. However, judging from the data there is hints that oligomerisation may play a role. Thus, can higher order structures of ZAP/Nuclease complexes on RNA only be found in the presence of TRIM25. Could the authors assess whether these oligomers of complexes exist, for example either by native page analyses or in vitro reconstitution of the complex on RNA? To show that E3 ligase activity is dispensable, Fig 6 could include mutants of TRIM25 that are RING defective i.e. C50S/C53S. In addition, could the authors discuss why an E3 ligase with RNA binding function is enhancing the function of such a complex, although its two major functions are not required?

Anti-viral complexes are usually highly spatially coordinated. While the authors show that binding (often) correlates with function, can this data be augmented by spatial analysis of key interactions e.g. using confocal microscopy (colocalization or proximity ligation assays). For example, this

would be beneficial in case where subcomplex binding is still detected but the function is absent (Fig. 2, the different ZAP isoforms, especially S, have varying activity which cannot be explained by co-factor binding alone, however complex assembly could be altered; Fig. 3 dCTD mutant vs TRIM25/ZAP)

The 'artificial' functional ZAP complex is intriguing and a nice proof of principle creating a minimal functional anti-viral factor. Can the authors envision any applications? Fig. 7b: Would co-transfection of TRIM25 further enhance the function of the fusion protein? The authors show that the SPRY domain of TRIM25 is critical for its enhancing function (Fig. 6g), why was this domain not included in the fusion constructs, and if so, does it do anything?

Minor comments

It would be helpful to have an overview figure that aggregates the information regarding binding and function in one figure panel.

Fig. 2b and c: It would be good if an additional panel e.g. area under curve, shows the aggregated activity over multiple concentrations, to help the reader compare the different variants.

Fig 6f and g: It would be great to compare the additional impact of TRIM25 in a more condensed overview as an additional figure panel.

Please include more details about the structure which was resolved in the results section (Currently: "To determine the molecular details of this interaction, we solved a 2.30-Å crystal structure of ZAP-X bound to KHNYNCTD") about the quality, method and reliability of the reconstruction and not only reference to the dataset. Please also deposit the raw data to e.g. PDB. In general the crystal structure could be highlighted a bit more in the abstract/results sections.

The model in Fig. 8 is very complex and could be moved to the supplement. Ideally, the mirrored text should also be removed.

Reviewer #3 (Remarks to the Author):

The antiviral host factor ZAP plays a critical role in viral RNA destruction and virus attenuation. Among the many ZAP cofactors, TRIM25 and KHNYN appear to be particularly important for ZAP-mediated antiviral activity. In the current work, the authors generated a robust reconstitution system to functionally investigate the requirements of TRIM25, KHNYN and N4BP1 (redundant

homolog of KHNYN) for ZAP antiviral activity directed against CpG-enriched viral genomes. In addition, the authors dissected the functional anatomy of the ZAP/TRIM25/KHNYN complex, determined a crystal structure of ZAP N-terminal domain in complex with KHNYN C-terminal domain, and designed artificial chimeric proteins that reconstituted the antiviral function of the ZAP/TRIM25/KHNYN complex. Overall, this is an informative and interesting work that adds to our understanding of the antiviral function of the ZAP-related complexes.

Questions:

1. When mapping the interaction domains between KHNYN and ZAP, it would be better to exclude the possibility that RNA might mediate some indirect contacts.

2. To further confirm the function of N4BP1 as a redundant homolog of KHNYN, the authors may want to provide more evidence for the ZAP/KHNYN-N4BP1 interactions, the nuclease activity of N4BP1, and the antiviral activity of a ZAP-N4BP1 chimeric protein.

3. Fig.2d, why is the binding strength of ZAP-L-Flag & ZAP-N-HA and ZAP-N-Flag & ZAP-L-HA so different?

4. Fig.2c, why does ZAP-X (RNA null) still show a CpG-imposed attenuation?

5. It would be better to include a drawing showing the differences between KHNYN-CTD and the previous structure in terms of connectivity, spatial arrangement, and length of the helices.

6. Too many typos!

--Page2, Abstract, "we could designed" should be "design";

--Page4, "and that that another" should delete a "that";

--Page4, "Fig. 1a e" should be "Fig. 1a-e";

--Page7, "A Z AP mutant" should be "A ZAP mutant";

--Page12, the last sentence should end with a period;

--Page12, "TRIM proteins that that" should delete a "that";

RESPONSES TO REVIEWER COMMENTS

Reviewer comments are pasted below along with our responses to them in blue typeface

Reviewer #1 (Remarks to the Author):

In this study, the authors investigated the functional and physical interactions of ZAP with KHNYN and TRIM25, two ZAP co-factors that have been previously reported to be important for the antiviral activity of ZAP with the underlying mechanisms being not very clear. They find that multiple domains of KHNYN contribute to its interaction with ZAP. Furthermore, they provide experimental evidence showing that KHNYN is an active RNA nuclease. They also map the TRIM25 determinants that enhanced ZAP activity. While these findings represent an important advance in understanding how the antiviral activity of ZAP is executed and regulated, two major concerns need to be addressed. First, the authors use HIV-1WT as a negative control in the studies of the antiviral activity of ZAP. However, multiple results in this manuscript showed that HIV-1WT was inhibited by ZAP although the inhibitory effect was not as dramatic as on HIV-1CG. A likely explanation of these results is that the sensitivity of HIV-1WT to ZAP is just moderate and enhanced by the introduction of CG dinucleotide in HIV-1CG. This notion needs to be clarified in the manuscript to avoid confusion. In addition, both HIV-1CG and EV71CG used in this study are artificial viruses. To validate the biological relevance of the main findings, the authors need to show that KHNYN and TRIM25 are important for the antiviral activity of ZAP against a naturally existing virus, say Sindbis virus. This would tell us whether the observations are specific to the artificial viruses. Second, the authors repeatedly used the term ZAP/KHNYN/TRIM25 complex. They need to provide experimental results to show the existence of the complex, say by co-IP assays. In summary, the findings reported in this manuscript are potentially important but some issues need to be addressed to support the main conclusions.

Specific comments:

1. Figure 1, as said above, the authors need to use a naturally existing virus to test the requirement of TRIM25 and KHNYN for the antiviral activity of ZAP in these cells. The authors could compare the replication kinetics of the virus in the different KO cell lines. This should be an easy experiment.

Response: The effect of TRIM25 and KHNYN on the replication of other viruses (e.g. Sindbis, JEV, MLV) has been documented in other papers (Yang et al (2022); PMID: 35800389; Zheng et al, (2017); PMID: 28202764; Li et al (2017); PMID: 28060952, Ficarelli et al (2019); PMID: 31284899)

2. In Figure 1m, HIV-1WT was also inhibited by about 10-fold. However, in Figure 1n, this inhibition was not observed, why?

Response: In Fig 1m, TRIM25 is absent, while in Fig 1n TRIM25 is present. One conclusion of the paper is that the presence of TRIM25 increases the specificity of ZAPs antiviral activity for CG-rich viruses, as well as the overall activity of ZAP. We concede that our model for explaining the differential levels of CpG specificity for the various reconstituted antiviral proteins/complexes in the various antiviral assays was not well articulated in the original manuscript and we have explained why we think TRIM25 and other complex components contribute to CpG specificity more clearly in the revised manuscript.

3. Figure 7b, the ZAP(RBD)-NYN fusion protein displayed robust antiviral activity against HIV-1WT, why? This needs to be better explained.

Response: We concede that our model for explaining the differential CpG specificity in the various antiviral assays was not well articulated in the original manuscript. Basically, we surmise that components of complexes other than the core monomeric ZAP RNA binding domain contribute to

CpG specificity. While the monomeric ZAP RNA binding domain itself exhibits a degree of specific binding to RNAs containing CpG dinucleotides (see our previous paper Meagher et al (2019) PMID: 31719195), the CpG binding pockets therein are embedded within a larger RNA binding surface which also has some ability to bind RNAs that lack CpG dinucleotides. The revised manuscript contains a more extensive discussion/argument, and a more clearly described model (shown in Fig 8 of the revised paper) Specifically, we propose that the interaction between ZAP and RNA is multivalent, with TRIM25 and ZAP mediated multimerization contributing to specificity for target RNAs that contain multiple CpG dinucleotides. While aspects of the model are somewhat speculative, they are consistent with the data, and we think are reasonable in the light of data presented in the paper.

4. Discussion: as said above, the authors need to show the existence of the complex.

Response: The paper contains numerous co-immunoprecipitation experiments showing that ZAP and KHNYN associate with each other, that ZAP and TRIM25 associate with each other, that TRIM25 and KHNYN associate with each other, that TRIM25 self-associates and that ZAP self-associates. We accept that these experiments do not formally demonstrate the existence of a *single* ternary complex simultaneously containing all three proteins and we have adjusted the title of the paper and the text accordingly. While it is, in our view, extremely likely that the ternary complex exists, we have edited the paper so as not to explicitly make this claim.

Minor:

1. In Figure 1a-h, HIV-1WT seems to replicate better in HeLa-Ctrl cells than in the other cells, especially the ZAP KO cells, any explanation?

Response: It is our experience that subclones of HeLa cells (and basically any cell line or primary cells), and different passage history exhibit clonal variation in their ability to support HIV-1 replication. What is important in these experiments is the difference between viruses that do (or do not) have elevated CpG dinucleotides.

2. The Input and CoIP blots are not aligned.

Response: This has been corrected in the revised manuscript

3. There are many grammar errors that need to be fixed. For example: in the Abstract, “we could designed artificial.....”

Response: This has been corrected in the revised manuscript

Reviewer #2 (Remarks to the Author):

The zinc finger antiviral protein (ZAP) is a well-known and broad acting restriction factor, that targets CpG islands in virus-derived RNA for degradation. To this end, ZAP recognizes the target RNA then recruits co-factors like the nuclease KHNYN or N4BP1. In addition, the E3 ligase TRIM25 is recruited to promote the function of the complex. However, how the subunits of this complex are coordinated was for far unknown.

In this manuscript, Bohn and colleagues systematically assesses the domains/functions required for the proper assembly of the complex as well as that both recruited nucleases work in a redundant manner. Furthermore, the functional interface of the complex is presented as a crystal structure. In addition, the authors construct a minimal functional complex containing of fused protein domains. Overall this is a well-written and systematic study which provides novel insight into a fascinating antiviral factor complex. The data largely supports the conclusions of the manuscript. I have just a

few major suggestions as well as a few minor comments to improve the current manuscript and clarify a few inconsistencies.

The big question is still what TRIM25 is doing in that complex to enhance its function since, ubiquitination seems to be not required and RNA binding function is dispensable. However, judging from the data there is hints that oligomerisation may play a role. Thus, can higher order structures of ZAP/Nuclease complexes on RNA only be found in the presence of TRIM25. Could the authors assess whether these oligomers of complexes exist, for example either by native page analyses or in vitro reconstitution of the complex on RNA?

Response: These are good suggestions, and we are obviously attempting these experiments as part of structural studies of protein complexes contain ZAP, TRIM25 and or KHNYN. However, the full-length proteins are quite poorly behaved in in vitro, and assembly of the complexes in vitro or in a form that would survive PAGE has thus far eluded us. The great deal of optimization work that would be required to complete these studies is (we think) beyond the scope of this study.

To show that E3 ligase activity is dispensable, Fig 6 could include mutants of TRIM25 that are RING defective i.e. C50S/C53S. In addition, could the authors discuss why an E3 ligase with RNA binding function is enhancing the function of such a complex, although its two major functions are not required?

Response: Residues C50 and C53 are zinc coordinating residues therefore their substitution is expected to unfold the RING domain. As such, the poor activity of a C50S/C53S mutant would not distinguish whether RING driven E3 ligase ligation activity or RING domain driven multimerization are important, as both of which would be abolished by unfolding the RING domain. Note that the 'CCSS' mutant shown in Fig S6a of the original manuscript is, in fact, a C50S/C53S mutant (we have changed the revised manuscript to clarify this). Importantly, an X-ray crystal structure of the TRIM25 RING domain in complex with itself, the E2 enzyme Ubc13 and ubiquitin has been determined, and amino acids required for each of these interactions mapped. This enabled us to precisely target mutations to determine which TRIM25 interactions/activities are required to support ZAP antiviral activity. We have pointed this out more explicitly in the revised manuscript. Fig S6 shows a number other mutants at the Ub-RING and the E2-RING interfaces (that block Ubiquitin conjugation activity) do not affect TRIM25 activity in the antiviral assay, while several mutants at the RING-RING interface do inhibit ZAP activity. To us this constitutes very strong evidence that RING driven multimerization is the key TRIM25 activity supporting ZAP function. We have included a more extensive discussion of the role of TRIM protein activities in their function in the revised manuscript.

Anti-viral complexes are usually highly spatially coordinated. While the authors show that binding (often) correlates with function, can this data be augmented by spatial analysis of key interactions e.g. using confocal microscopy (colocalization or proximity ligation assays). For example, this would be beneficial in case where subcomplex binding is still detected but the function is absent (Fig. 2, the different ZAP isoforms, especially S, have varying activity which cannot be explained by co-factor binding alone, however complex assembly could be altered; Fig. 3 dCTD mutant vs TRIM25/ZAP)

Response: it is our experience that ZAP is diffusely distributed across the entire cytoplasm. Colocalization assays have thus not been informative.

The 'artificial' functional ZAP complex is intriguing and a nice proof of principle creating a minimal functional anti-viral factor. Can the authors envision any applications?

Response: The fusion of sequence specific RNA binding protein to a nuclease could have many applications! We have included a brief discussion of potential applications in the discussion of the revised manuscript.

Fig. 7b: Would co-transfection of TRIM25 further enhance the function of the fusion protein?

Response: In fact, the experiment in Fig. 7b already shows the activity of the ZAP-NYN fusion in the presence and absence of TRIM25. In fact, TRIM25 has no effect in this context, this was explicitly stated in the original manuscript and is also in the revised version.

The authors show that the SPRY domain of TRIM25 is critical for its enhancing function (Fig. 6g), why was this domain not included in the fusion constructs, and if so, does it do anything?

Response: The point of this experiment was to determine whether the SPRY domain becomes dispensable if the ZAP RBD is fused directly to TRIM25. Indeed, it does, as indicated by the fact that the TRIM25(del Spry)-ZAP RBD fusion is as active as the full length TRIM25 and ZAP RBD expressed separately (Fig 7e). In contrast, the SPRY domain is crucial for activity if TRIM25 and ZAP RBD expressed separately (Fig. 6c).

Minor comments

It would be helpful to have an overview figure that aggregates the information regarding binding and function in one figure panel.

Response: This is a good suggestion, and we have substituted the existing summary figure (Fig 8) with a new version that aggregates the information and proposes a model that describes the interactions and function for the various components studied herein.

Fig. 2b and c: It would be good if an additional panel e.g. area under curve, shows the aggregated activity over multiple concentrations, to help the reader compare the different variants.

Response: This is tricky to do without obscuring some details that are important (particularly those pertaining to varying effects on CpG specificity that are different at different points in the dose response for the various truncated ZAP proteins). While we appreciate the reviewers' desire for brevity, we think that these points are important to communicate.

Fig 6f and g: It would be great to compare the additional impact of TRIM25 in a more condensed overview as an additional figure panel.

Response: We have included an additional figure panel (Fig 6) that includes summary data.

Please include more details about the structure which was resolved in the results section (Currently: "To determine the molecular details of this interaction, we solved a 2.30-Å crystal structure of ZAP-X bound to KHNYNCTD") about the quality, method and reliability of the reconstruction and not only reference to the dataset. Please also deposit the raw data to e.g. PDB. In general the crystal structure could be highlighted a bit more in the abstract/results sections.

Response: As indicated in Table S2, the diffraction data and atomic coordinates for the crystal structure of the ZAP_{RBD}:KHNYN_{CTD} complex are deposited in the PDB with accession code 9BGL. These will be publicly available when the manuscript is published. The quality and reliability of the diffraction data and the refined structure were validated using standard software and are summarized in Table S2. At the editor's request, we provided the full PDB validation report after the initial manuscript submission.

The abstract has an additional sentence about the structure (underlined).

“The ZAP N-terminal RNA binding domain acts as the minimal essential core of the antiviral complex, and we present a crystal structure of this ZAP domain that reveals contacts with the functionally required KHNYN C-terminal domain. These contacts are remote from the ZAP binding site for CpG and would not interfere with RNA binding.”

The Results section is updated with:

“In our crystal structure, the KHNYN_{CTD} has the three-helix bundle structure of other CUE ubiquitin binding domains (^{1,2}PDB 2EJS), but in the previous structure, the connectivity, spatial arrangement and length of the helices differ.”

The model in Fig. 8 is very complex and could be moved to the supplement. Ideally, the mirrored text should also be removed.

Response: We have substituted the existing summary figure (Fig. 8), that we concede ‘missed the mark’ in terms of communicating an effective summary of the results, with a new version that aggregates the information and proposes a model that describes the interactions and function for the various components studied herein.

Reviewer #3 (Remarks to the Author):

The antiviral host factor ZAP plays a critical role in viral RNA destruction and virus attenuation. Among the many ZAP cofactors, TRIM25 and KHNYN appear to be particularly important for ZAP-mediated antiviral activity. In the current work, the authors generated a robust reconstitution system to functionally investigate the requirements of TRIM25, KHNYN and N4BP1 (redundant homolog of KHNYN) for ZAP antiviral activity directed against CpG-enriched viral genomes. In addition, the authors dissected the functional anatomy of the ZAP/TRIM25/KHNYN complex, determined a crystal structure of ZAP N-terminal domain in complex with KHNYN C-terminal domain, and designed artificial chimeric proteins that reconstituted the antiviral function of the ZAP/TRIM25/KHNYN complex. Overall, this is an informative and interesting work that adds to our understanding of the antiviral function of the ZAP-related complexes.

Questions:

1. When mapping the interaction domains between KHNYN and ZAP, it would be better to exclude the possibility that RNA might mediate some indirect contacts.

Response: All co-immunoprecipitation experiments were done with Ribonuclease A included during cell lysis. This was clearly stated in the methods, section but we have added mention of this in the results section for additional clarity.

2. To further confirm the function of N4BP1 as a redundant homolog of KHNYN, the authors may want to provide more evidence for the ZAP/KHNYN-N4BP1 interactions, the nuclease activity of N4BP1, and the antiviral activity of a ZAP-N4BP1 chimeric protein.

Response: N4BP1 was difficult to express, particularly in its catalytically active form. In many experiments our attempts to express it in intact or truncated forms resulted in undetectable levels of the protein by western blotting.

3. Fig.2d, why is the binding strength of ZAP-L-Flag & ZAP-N-HA and ZAP-N-Flag & ZAP-L-HA so different?

Response: We suspect that this is because the ZAP-L-Flag and ZAP-N-Flag co-IP “bait” proteins were stably expressed at significantly lower levels than the transiently transfected ZAP-N-HA and ZAP-L-HA “prey” proteins. Then - if ZAP-L homodimers assemble more efficiently than ZAP-N/ZAP-L

heterodimers or ZAP-N/ZAP-N homodimers, then an excess of ZAP-N would be required to form abundant ZAP-N/ZAP-L heterodimers if both ZAP-L and ZAP-N proteins are present. We have included a brief discussion of this point in the revised manuscript.

4. Fig.2c, why does ZAP-X (RNA null) still show a CpG-imposed attenuation?

Response: The ZAP-X (RNA null) mutant encodes mutations at positions (R74A, R75A, K76A). These substitutions are in the larger RNA binding site on ZAP but not at residues that specifically contact CpG dinucleotides. (Meagher et al (PMID: 31719195)). The residual antiviral activity (which is only a few % of that of the WT protein) may therefore be specific for CpG-enriched HIV-1.

5. It would be better to include a drawing showing the differences between KHNYN-CTD and the previous structure in terms of connectivity, spatial arrangement, and length of the helices.

Response: A new panel is included in Figure S5 to compare our structure of the KHNYN CTD, the previously published solution structure of the same protein (PDB 2N5M), and the structures of one close homolog (PDB 8T48) and two distant homologs (PDB 2EJS and 6H3A). Over a considerable range of sequence identities (13-40%), our crystal structure is similar to the close and distant homologs (crystal structures 8T48 & 6H3A, NMR structure 2EJS). All of the similar structures are radically different from the published solution structure of the KHNYN CTD (PDB 2N5M), which we conclude is incorrect. While it is important to be clear about this, we do not wish to over-emphasize others' mistakes, so included only 2 sentences in the main text. After our study is accepted for publication, we will share our structure with the authors of the earlier KHNYN CTD study.

6. Too many typos!

--Page2, Abstract, "we could designed" should be "design";

--Page4, "and that that another" should delete a "that";

--Page4, "Fig. 1a e" should be "Fig. 1a-e";

--Page7, "A Z AP mutant" should be "A ZAP mutant";

--Page12, the last sentence should end with a period;

--Page12, "TRIM proteins that that" should delete a "that";

Response: We have corrected these typos in revised manuscript

REVIEWER COMMENTS

Reviewer #1 (Remarks to the Author):

My concerns have been addressed.

Reviewer #2 (Remarks to the Author):

I consider the overall concept and the data still very interesting and novel. It is a tad disappointing that the authors chose to experimentally address practically none of my suggestions. Thus, I cannot really say that the manuscript improved significantly during this revision. While most of my minor comments were sufficiently addressed, my original assessment that “the data largely supports the conclusions of the manuscript. I have just a few major suggestions” is still applicable for this revised version.

In particular, higher order complexes of TRIM25/ZAP/other co-factors could be shown using proximity ligation assays, differentially tagged IPs, native PAGEs etc. I do not think that any of these assays are beyond the scope of the study as claimed by the author. However, I do agree that in depth in vitro crystal structural studies would most likely be beyond the scope (according to my opinion).

I was wondering whether there is a mutant of the RING domain, which abrogates multimerization, which may answer the question whether multimerization, and not ubiquitination is required for activity.

During infection the ZAP/TRIM25 often localises to stress granules. Co-localisations should have been attempted in stimulated or infected cells. Proximity ligation assays would suggest a spatial proximity despite a diffuse distribution across the cytoplasm.

Reviewer #3 (Remarks to the Author):

The authors have addressed my concerns.

RESPONSES TO REVIEWER COMMENTS

Reviewer comments are pasted below along with our responses to them in blue typeface

Reviewer #2 (Remarks to the Author):

I consider the overall concept and the data still very interesting and novel. It is a tad disappointing that the authors chose to experimentally address practically none of my suggestions. Thus, I cannot really say that the manuscript improved significantly during this revision. While most of my minor comments were sufficiently addressed, my original assessment that “the data largely supports the conclusions of the manuscript. I have just a few major suggestions” is still applicable for this revised version.

In particular, higher order complexes of TRIM25/ZAP/other co-factors could be shown using proximity ligation assays, differentially tagged IPs, native PAGEs etc. I do not think that any of these assays are beyond the scope of the study as claimed by the author. However, I do agree that in depth in vitro crystal structural studies would most likely be beyond the scope (according to my opinion).

Response: In the second revision we have included an additional experiment (shown in Fig. S3c of the revised manuscript) in which both endogenous ZAP-L and endogenous TRIM25 are simultaneously coimmunoprecipitated using tagged KHNYN. We have included an additional paragraph of text (page 9) to describe this result. This suggests that a ternary complex containing all three proteins can form.

During infection the ZAP/TRIM25 often localises to stress granules. Co-localisations should have been attempted in stimulated or infected cells. Proximity ligation assays would suggest a spatial proximity despite a diffuse distribution across the cytoplasm.

Response: we have previously looked into the localization of ZAP and TRIM25 in unpublished work. While ZAP/TRIM25 can both relocate to stress granules in some situations (e.g. oxidative stress), the relocation of each protein to stress granules is not dependent on the other (e.g. TRIM25 relocates to stress granules upon oxidative stress even in the absence of ZAP)- thus their colocalization in stress granules is not informative as to whether the two proteins interact. Moreover, we have found that stress granules clearly do NOT form in cells infected with HIV-1_{CG}, even under conditions where the ZAP antiviral complex is exerting maximal inhibition of virus replication. Thus, localization to stress granules cannot be important for antiviral activity.

Additionally, we have tried proximity ligation assays in other contexts and simply do not regard this assay as a reliable reporter of protein interactions. Specifically, in our hands pairs of proteins that are not expected to interact with each other can give positive signals. Our experience is mirrored in a recent preprint from Alsemarz et al. (doi: <https://doi.org/10.1101/411355>) “Limited significance of the in situ proximity ligation assay” (abstract pasted below).

Limited significance of the in situ proximity ligation assay

Azam Alsemarz^{1,2}, Paul Lasko² and François Fagotto^{1,2}
1CRBM, University of Montpellier and CNRS, Montpellier 34293, France and 2Dept. of Biology, McGill University, Montreal, QC, Canada H3A1B1. Correspondence to François Fagotto, email: francois.fagotto@crbm.cnrs.fr

Summary

In situ proximity ligation assay (isPLA) is an increasingly popular technique that aims at detecting the close proximity of two molecules in fixed samples using two primary antibodies. The maximal distance between the antibodies required for producing a signal is 40 nm, which is lower than optical

resolution and approaches the macromolecular scale. Therefore, isPLA may provide refined positional information, and is commonly used as supporting evidence for direct or indirect protein-protein interaction. However, we show here that this method is inherently prone to false interpretations, yielding positive and seemingly 'specific' signals even for totally unrelated antigens. We discuss the difficulty to produce adequate specificity controls. We conclude that isPLA data should be considered with extreme caution.

I was wondering whether there is a mutant of the RING domain, which abrogates multimerization, which may answer the question whether multimerization, and not ubiquitination is required for activity.

Response: While the TRIM25 RING domain crystalizes as a dimer ((Sanchez et al [reference 24]) RING domain dimerization is not possible to assay directly, as the isolated RING domain purifies as a monomer in solution. Moreover, because ubiquitin ligase activity requires RING domain dimerization, it is only partly possible to genetically separate these two activities. However, our analysis of an extensive series of RING domain mutations (Fig. 6c and Fig. S6a) was guided by the published structure of the dimeric TRIM25 RING domain in a complex with an E2 enzyme Ubc13 and Ubiquitin. This enabled selective mutation of amino acids (i) at the RING-RING interface, (ii) the RING:Ubc13 interface and (iii) RING:Ubiquitin interfaces. Crucially, this set included mutations that retain an intact RING:RING interface but been demonstrated to lack ubiquitin ligase activity (e.g. K65A, N71D and C50S/C53S) or lack the crucial ubiquitin acceptor site (K117R) (Sanchez et al [reference 24] and Inn et al [reference 35]), These mutants retain activity in the antiviral assay. Thus, TRIM25 ubiquitin ligase activity cannot be essential. In contrast, mutations at the RING:RING interface generally inhibited TRIM25 activity in the reconstitution assay, indicating the integrity of the RING:RING interface, and by inference RING driven multimerization, is important for antiviral activity.

REVIEWERS' COMMENTS

Reviewer #2 (Remarks to the Author):

The authors have addressed all my concerns.